

# Impact of rain-on-snow events on snowpack structure and runoff under a boreal canopy

Benjamin Bouchard[1,2,3], Daniel F. Nadeau[1,2], Florent Domine[3,4,5], Nander Wever[6,7], Adrien Michel[6,7,8], Michael Lehning[6,7], Pierre-Erik Isabelle[1,2]

[1]Department of Civil and Water Engineering, Université Laval, Quebec City, G1V 0A6, Canada
[2]CentrEau – Water Research Centre, Université Laval, Quebec City, G1V 0A6, Canada
[3]Centre d'Études Nordiques, Université Laval, Quebec City, G1V 0A6, Canada
[4]Department of Chemistry, Université Laval, Quebec City, G1V 0A6, Canada
[5]Takuvik Joint International Laboratory, Université Laval (Canada) and CNRS-INSU (France), Quebec City, G1V 0A6,
Canada
[6] WSL Institute for Snow and Avalanche Research (SLF), 7260 Davos Dorf, Switzerland
[7] École Polytechnique Fédérale de Lausanne (EPFL), School of Architecture, Civil and Environmental Engineering,
Lausanne, Switzerland
[8] Federal Office of Meteorology and Climatology, MeteoSwiss, Geneva, Switzerland

*Correspondence to*: Benjamin Bouchard (benjamin.bouchard.1@ulaval.ca)

**Abstract.** Rain-on-snow events can cause severe flooding in snow–dominated regions. These are expected to become more frequent in the future as climate change shifts the precipitation from snowfall to rainfall. However, little is known about how winter rainfall interacts with an evergreen canopy and affects the underlying snowpack. In this study, we document 5 years of rain-on-snow events and snowpack observations at two boreal forested sites of eastern Canada. Our observations show that

rain-on-snow events over a boreal canopy leads to the formation of melt–freeze layers as rainwater refreezes at the surface of the sub–canopy snowpack. They also generate frozen percolation channels, suggesting that preferential flow is favored in the sub–canopy snowpack during rain-on-snow events. We then used the multi–layer snow model SNOWPACK to simulate the sub–canopy snowpack at both sites. Although SNOWPACK performs reasonably well in reproducing snow height (RMSE = 17.3 cm), snow surface temperature (RMSE = 1.0°C), and density profiles (agreement score = 0.79), its performance declines

when it comes to simulating snowpack stratigraphy, as it fails to reproduce many of the observed melt–freeze layers. To correct for this, we implemented a densification function of the intercepted snow in the canopy module of SNOWPACK. This new feature allows 27 of the 32 observed melt–freeze layers induced by rain-on-snow events to be formed by the model, instead of only 18 with the original canopy module. This new model development also delays and reduces the snowpack runoff. Indeed, it triggers the unloading of dense unloaded snow layers with small rounded grains, which in turn produces fine–over–coarse

transitions that limit percolation and favor refreezing. Our results show that the boreal vegetation modulates the sub–canopy snowpack structure and runoff from rain-on-snow events. Overall, this study highlights the need for canopy snow properties measurements to improve hydrological models in forested snow–covered regions.



## 1 Introduction

There are several definitions of a rain-on-snow (ROS) event (Brandt et al., 2022). Simply put, a ROS occurs when liquid
precipitation falls on an existing snowpack (McCabe et al., 2007). As simple as the definition of a ROS may be, these can have
major consequences if the right conditions are met (Wayand et al., 2015). Historically, some winter ROS events have caused
severe flooding and damage in North America, Europe, and many other regions (Marks et al., 1998; Haleakala et al., 2023;
Rössler et al., 2014). The risk of flooding from ROS events is controlled not only by atmospheric and soil conditions (Haleakala
et al., 2023; Zaqout et al., 2023), but also by the energy balance and structure of the snowpack. A warm and thin snowpack
favors a greater contribution of snowmelt to flooding (Jennings et al., 2018; Brandt et al., 2022), while a cold and thick snow
cover limits the occurrence of snowmelt (Trubilowicz and Moore, 2017). In cold and stratified snowpacks, preferential flow
tends to occur, causing rapid runoff during ROS events (Avanzi et al., 2016; Würzer et al., 2017). In contrast, snowpack runoff
is slowed by ice layers and melt–freeze crusts within the snowpack that impede water percolation (Webb et al., 2018a; Würzer
et al., 2016; Eiriksson et al., 2013). In the Northern Hemisphere, ROS are expected to become more intense with warmer
winters in response to a shift from solid to liquid precipitation, increasing the risk of flooding (IPCC, 2022; Musselman et al.,
2018).

The relationship between the presence of a forest and runoff from ROS remains unclear, as only a few studies have explored
this topic (Brandt et al., 2022). Beaudry and Golding (1983) and Berris and Harr (1987) found that tall and dense coniferous
forests in western North-America experienced reduced runoff during ROS events when compared to open sites. In contrast,
Berg et al. (1991) and Garvelmann et al. (2015) found that vegetation has a small impact on runoff following a ROS in mature
and dense mixed forests of the western United States and central Europe, respectively. None of these studies have examined
the role of the sub–canopy snowpack structure during ROS, despite its strong influence on water transport and snowpack
runoff.

Vegetation heavily influences the evolution of snowpack structure in snow–dominated regions. During winter, evergreen
canopies intercept a substantial fraction of the solid precipitation (Hedstrom and Pomeroy, 1998; Pomeroy et al., 2002),
limiting snow accumulation underneath (Varhola et al., 2010; Parajuli et al., 2020). A thin snowpack below the canopy favors
the diffusion of water vapor among snow layers, leading to gradient metamorphism and grain growth (Colbeck, 1983). Larger
grains increase the hydraulic conductivity of the snowpack (Bouchard et al., 2022). Canopy snow unloading, meltwater
dripping, and accumulation of vegetation debris enhance the sub–canopy snowpack heterogeneity, thereby increasing the
likelihood of preferential flow under the trees and the formation of melt–freeze layers (Bründl et al., 1999; Teich et al., 2019;
Bouchard et al., 2022). Since forest canopies influence snowpack structure and may impact snowpack runoff from ROS, it
would be logical to link snowpack structure to ROS runoff, but studies are needed to confirm this.

Detailed snow models can help capture the complex dynamics of water flow through the snowpack, and aid in the interpretation
of field observations. SNOWPACK (Lehning et al., 2002a) is a multi–layer, one–dimensional snow model that can solve
Richards' equation to describe water percolation from preferential flow within the snow cover (Wever et al., 2014; Wever et



al., 2016). SNOWPACK uses a big–leaf approach to represent the canopy (Lehning et al., 2006). Recently, a two–layer canopy module with thermal inertia has been implemented, which allows better representation of the diurnal variation of canopy temperature and heat exchanges with the sub–canopy snowpack (Gouttevin et al., 2015). SNOWPACK has been evaluated in forested environments with regard to snowpack energy balance (Gouttevin et al., 2015; Todt et al., 2018), snow accumulation

(Rutter et al., 2009; Gouttevin et al., 2015), and snow–related climate feedbacks (Krinner et al., 2018). However, only Rasmus et al. (2007) and Kontu et al. (2017) used detailed snow profile observations to validate SNOWPACK in the boreal forest of Finland. In both studies, the authors found a reasonable agreement between the model and the observations, but their model validation was limited to open forest clearings, without consideration for snow under the trees. Further evaluation of SNOWPACK to simulate the sub–canopy snow cover in northern regions is therefore in order.

SNOWPACK, as any other snow model, tends to underperform in forests when compared to open areas (Rutter et al., 2009). This is partly due to limited field data and interception parameterizations derived from a few observational studies that are difficult to generalize to other climates (Lundquist et al., 2021). In SNOWPACK, the interception scheme is based on experiments by Schmidt and Gluns (1991) and Hedstrom and Pomeroy (1998) conducted in a continental climate of the western United States and Canada. In this parameterization, the intercepted fraction of the precipitation is stored in the canopy until

the temperature dependent maximum capacity is reached (Pomeroy et al., 1998). The intercepted snow is then unloaded when the canopy storage exceeds its maximum capacity. The unloaded snow is considered "fresh snowfall", with properties defined as a function of meteorological variables such as air and snow surface temperature, relative humidity, and wind speed at the time of the unloading event (Lehning et al., 2002b). However, snow can remain in the canopy for up to several weeks (MacDonald, 2010; Lumbrazo et al., 2022) and undergo metamorphism before being unloaded. This can lead to inaccurate

simulations of the physical properties and stratigraphy of the sub–canopy snowpack after an unloading event, which in turn could alter the simulated downward water flow from a ROS event. Moreover, the high spatial heterogeneity of the unloading mechanisms makes it even more difficult to accurately simulate the sub–canopy snowpack using a one–dimensional snow model (Vincent et al., 2018).

In summary, this study aims to help fill the research gap on ROS events in the boreal forest. The combination of detailed field

observations and modelling of the sub–canopy snowpack properties is the ideal setup to gain insight on these events, which are expected to become more frequent. The first objective of this study is to use field observations to document the ROS–induced alterations of sub–canopy snowpack structure, with a focus on melt–freeze snow layers and preferential flow channels. This objective builds on the work of Bouchard et al. (2022), who found that the sub–canopy snowpack is highly heterogeneous and has high permeability, which would facilitate downward water flow. The second objective is to specifically evaluate the

performance of SNOWPACK underneath a boreal canopy. Finally, in a third objective, we aim to assess the impact of a time–based densification of intercepted snow on the sub–canopy snowpack structure and ROS–induced runoff. Overall, this work allows for an improved understanding of the relationship between snow–forest processes, snowpack structure and runoff during ROS events.



## 2 Observational data

### 2.1 Study sites

In this study, we compare two sites located in the boreal forest of eastern Canada where meteorological variables are continuously measured above the canopy. The Montmorency Forest (MF) is the main site, as it is more thoroughly instrumented and readily accessible in winter. The Bernard River Valley (BRV) is located some 700 km northeast of MF and requires a full day's drive and an hour ski–in to reach in winter. This site, which covers a different bioclimatic area than MF and where less data has been collected, is therefore used primarily to validate conclusions drawn from the main site.

### 2.1.1 Montmorency Forest

MF (47°17′18″ N; 71°10′05″ W) is located in the province of Quebec, Canada, at the southern edge of the boreal forest (Fig. 1a). The site receives an average (1991–2020) total annual precipitation of 1504 mm, of which about 40% falls as snow (station no. 7042395 Foret Montmorency https://climate.weather.gc.ca/climate_normals/index_e.html). The mean 2 m air temperature in December, January, and February (DJF) is –13.2°C.

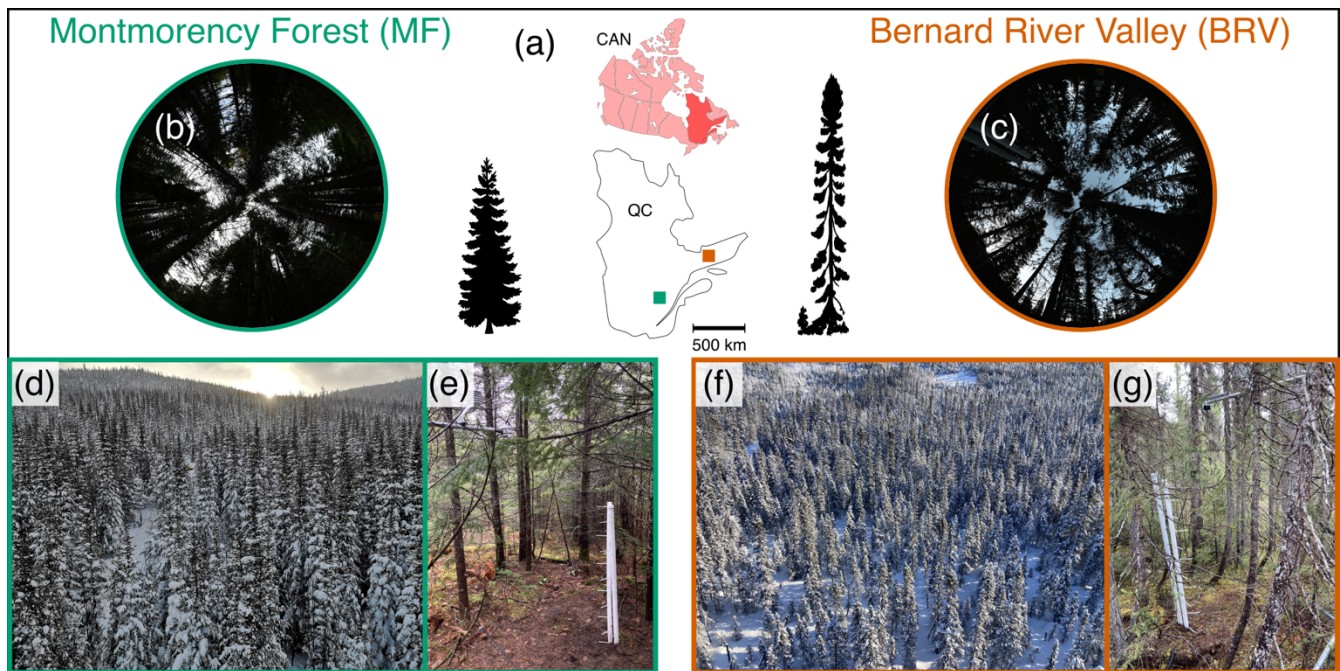

**Figure 1: Study sites. (a) Location in the province of Quebec (QC), Canada. (b–c) Hemispherical image of sub–canopy locations at both sites along with an artistic depiction of the dominant tree species at each site (balsam fir – MF; black spruce – BRV). (d–f) Photos taken from the meteorological towers showing the forest canopy. (e–g) Sub–canopy monitoring stations showing the thermistors array, snow height sensor and air temperature–relative humidity probe at both sites.**

The field campaign was conducted on a 12° northeast–facing slope at an elevation of ≈ 850 m above mean sea level (AMSL) within MF. The site is a juvenile balsam fir (*Abies balsamea*) – white birch (*Betula papyrifera*) stand that naturally regrew



after harvesting operations in 1993–1994 (Guillemette et al., 2005). The canopy height is between 7 and 12 m, with an average height of 9.2 m when estimated in the summer of 2019. The soil is a sandy loam overlaid by a 7 cm layer of organic material.

The average leaf area index (LAI) of sub–canopy locations at MF is $4.8 \pm 1.6$ m$^2$ m$^{-2}$. The LAI was derived from the analysis of 26 hemispherical canopy images using the method from van Gardingen et al. (1999), designed for highly clumped canopies such as dense coniferous stands. In this approach, the image is separated into $90 \times 1°$ zenith rings and the LAI is calculated from the non–linear estimation of the log–averaged gap fraction of each ring. In the summer of 2015, a 15 m flux tower was erected on site to measure hydrometeorological variables (Isabelle et al., 2018). In the fall of 2020, the tower and the

instruments were raised to a height of 20 m. Air temperature ($T_a$) and relative humidity ($RH$) were both measured using a HC2S3 probe (Rotronic). Incoming and outgoing shortwave ($SWR\downarrow$ and $SWR\uparrow$) and longwave radiation ($LWR\downarrow$ and $LWR\uparrow$) were monitored with a CNR4 radiometer (Kipp & Zonen) inclined so radiation was measured perpendicular to the slope. Wind speed ($WS$) was measured using a 2–D anemometer (R.M. Young – model 05103). Table 1 shows the measurement height of each meteorological variable at MF before and after raising the tower in October 2020. All variables are sampled every minute

and averaged to the half–hour.

Finally, a double fence automatic reference (DFAR) for liquid and solid precipitation ($P$) is located in an open clearing about 4 km northeast of the study site at $\approx 670$ m AMSL (Pierre et al., 2019). We assume that $P$ measured by the DFAR is representative of the precipitation received at the experimental site despite the distance and the elevation difference.

| Variable | MF – 2018–2020 (m) | MF – 2020–2023 (m) | BRV – 2019–2023 (m) |
|---|---|---|---|
| Air temperature | 15.24 | 18.29 | 24.00 |
| Relative humidity | 15.24 | 18.29 | 24.00 |
| All–wave radiation | 14.02 | 18.59 | 24.00 |
| Wind speed | 14.63 | 14.63 | 25.00 |

**Table 1: Measurement height from the ground of meteorological variables at Montmorency Forest (MF) and Bernard River Valley**
**(BRV)**

### 2.1.2 Bernard River Valley

The BRV site (50°54′36″ N; 63°22′48″ W ) receives 1077 mm of precipitation annually and the DJF temperature is –12.5°C, according to the 1991–2020 climate averages from the nearest federal weather station, located about 200 km southwest from the site (station no. 7047914 Sept-Iles https://climate.weather.gc.ca/climate_normals/index_e.html). The valley has a mean

elevation of $\approx 250$ m AMSL and is surrounded by plateaus 200 m higher in elevation. BRV is a site dominated by black spruce (*Picea mariana*), with trees between 12 and 18 m high, with an average height of 15 m, when estimated in the spring of 2021. The soil is composed of silty loam mineral soil under an organic layer of 17 cm. Compared to MF, the forest at BRV is rather sparse (Figs. 1c and 1f). We estimated an LAI of $1.6 \pm 0.7$ m$^2$ m$^{-2}$ from 55 LAI-2200C (Li-COR) measurements, also taken in the spring of 2021. A 25 m flux tower is used to measure the same meteorological variables as at the MF site. Air temperature

and relative humidity were measured using an HMP device, all–wave radiation was measured by a CNR4 and wind speed was obtained from 2–D anemometer measurements. The same sampling frequency and averaging time step as at MF are used and the height of the instrument is presented in Table 1.



## 2.2 Snow observations

Snow observations cover the period of 2018 to 2023 at MF (five winters) and 2019 to 2023 at BRV (four winters). Sub–canopy
monitoring stations were installed approximately 25 meters from the meteorological tower at each site prior to the first field
campaign. Continuous automated measurements were complemented by recurrent snow pit observations. These were
conducted at sub–canopy sites with similar characteristics to the monitoring station and within a radius of 100 m of the tower
at both sites.

### 2.2.1 Snow monitoring stations

At the MF site, snow height was monitored with an SR50 ultrasonic sensor (Campbell Scientific) in the winter 2018–19, and
with a Judd communication snow height sensor from 2019 to 2023. Snow surface temperature ($T_{surf}$) was monitored with an
SI-111 infrared radiometer (Apogee Instruments) in 2018–19 and from 2020 to 2023. Snow height and $T_{surf}$ measurements
were recorded at an hourly time step on a CR10X data logger (Campbell Scientific) from mid–October to mid–June. During
winters of 2020–21, 2021–22 and 2022–23, we also monitored snowpack temperature every 15 cm from the ground level with
Pt-1000 thermistors (Schneider Electric) and $T_a$ at 2.5 meters high using an HMP probe (Figs. 1e and 1g). A more detailed
description of the automatic stations can be found in Bouchard et al. (2023).

At the BRV sites, snow height was measured using a SR50 ultrasonic sensor from 2019 to 2021, and then with a Judd
communication snow height sensor from 2021 to 2023. We also monitored snowpack temperature every 15 cm using Pt-1000
thermistors in a setup similar to that described for MF. $T_{surf}$ was not measured at BRV.

At both the MF and BRV sites, we installed time–lapse cameras on trees at a height of 2.5 meters and facing the canopy. This
setup allowed us to detect the presence of snow in the canopy and to identify the phase of the precipitation as in Floyd and
Weiler (2008).

### 2.2.2 Snow pit measurements

A total of 48 snow pits were dug under the canopy during the 2018–2023 study period (42 at MF and 6 at BRV). Of this
number, 26 snow pits were dug at MF during the winter of 2018–19 (Bouchard et al., 2022). All snow pits included a visual
assessment of the snowpack stratigraphy, with detailed identification of ice and melt–freeze layers within the snowpack. Grain
types were identified using a magnifying glass and a millimetric grid by the same observer throughout the study period. Snow
pits also included a measurement of the vertical profile of snow density, except for the last snow pit of winter 2022–23 at MF.
Snow density was measured every 3 to 5 cm with a Snow-Hydro 100 cm³ box with a ±10% accuracy (Conger and McClung,
175   2009).





## 2.3 Soil observations

Soil profile characterizations were performed in the fall of 2020 (BRV) and the summer of 2021 (MF). We measured a vertical profile of temperature using a Pt-1000 thermistor (Greinsinger), soil thermal conductivity using a TP02 heated needle probe (Hukselflux), soil volumetric water content and density by gravimetric analysis, along with a soil texture identification through
the profile. Soil properties were measured up to depths of 100 cm at MF and 82 cm at BRV. More details on the observed soil characterization are provided in the Supplementary Material (Figs. S1 and S2).

## 2.4 Rain-on-snow events

In this study, we define a ROS event as at least 3 mm of rain amounted over 12 hours or longer while a minimum of 3 cm of snow on the ground is observed. It is one of the many definitions of a ROS event in the literature (Brandt et al., 2022) that we
chose for its simplicity in the absence of runoff measurements. Note that we focus on ROS events that occur between November and March exclusively.

## 3 Numerical modeling

We used the one–dimensional physically based snow cover model SNOWPACK (version 3.6.0) to simulate the sub–canopy snowpack from 2018 to 2023 in the Montmorency Forest and from 2019 to 2023 in the Bernard River Valley. The model
accounts for the processes that drive snow metamorphism and provides a complete representation of snow microstructure, thermal profile, water transport, snow settlement, and mass and energy balance. The full description of the model can be found in Bartelt and Lehning (2002), Lehning et al. (2002a), Lehning et al. (2002b) and in the online documentation (https://snowpack.slf.ch/doc-release/html/general.html). Simulations were run at a 15 min calculation time step. Contents of SNOWAPCK initialization files are presented in Appendix A.

### 3.1 Water transport scheme

The water transport scheme in SNOWPACK includes two options: a simple bucket model approach and a solution of Richards' equation (Wever et al., 2014). The latter approach allows for better simulation of snowpack runoff volume and timing, particularly in the early melt season and on sub–daily time scales (Wever et al., 2014; Wever et al., 2015). Wever et al. (2016) further implemented a description of preferential flow using a dual–domain approach, where water is transferred from the
preferential to the matrix flow domain when it encounters a layer transition, to simulate fine–over–coarse snow layers (Katsushima et al., 2013; Avanzi et al., 2016). In the model, this is conceptually characterized by a saturation threshold of the preferential flow domain ($\theta_{TH}$; $0-1$), which controls the movement of water from preferential to matrix flow. Phase change is only possible in the matrix flow domain. An ice reservoir parameterization was further developed by Quéno et al. (2020) to better capture the formation of continuous ice layers from discontinuous and growing ice lenses. This development improved
the formation of ice and melt–freeze layers and reduced the number of simulated ice layers that were not observed.





In the simulations, we set $\theta_{TH}$ to 0.35 at both sites. Although Wever et al. (2016) used $\theta_{TH} = 0.08$ in their study for an alpine snowpack, we used a larger threshold since the sub–canopy snowpack is more likely to promote preferential flow (Teich et al., 2019). A two year sensitivity analysis performed with SNOWPACK at an open site, i.e. without vegetation effects, shows that low $\theta_{TH}$ leads to a deep wetting of the snowpack instead of melt–freeze layer formation (not shown). Moreover, the snow

water equivalent appeared to be almost unsensitive to changes of $\theta_{TH}$. We took the geometric mean for the calculation of the hydraulic conductivity at the interface nodes between snow layers based on the analysis from Wever et al. (2015).

## 3.2 SNOWPACK canopy module

The canopy module of SNOWPACK relies on a two–layer canopy scheme as detailed by Gouttevin et al. (2015). The mass balance in the canopy depends on snow interception, canopy evaporation and sublimation, and unloading. The interception

rate is calculated from the canopy storage saturation, which varies with intercepted snow density ($\rho_{s,int}$) and LAI. The parameterization of $\rho_{s,int}$ is the same as for new snow (Lehning et al., 2002a). Evaporation and sublimation are calculated as part of the two–layer energy balance and added to the mass balance at the end of each time step. A complete description of the energy balance parametrization can be found in Sects. 2.4 and 2.5 of Gouttevin et al. (2015). Unloading occurs when canopy storage exceeds the maximum storage capacity and is combined with precipitation to contribute to the formation of a new

snow layer on top of the snowpack. A more detailed description of interception and unloading parameterization can be found in Appendix B.

### 3.2.1 Intercepted snow densification

As part of our third research objective, we implemented an age–based densification function in SNOWPACK to account for the evolution of the intercepted snow density (Fig. 2). From now on, we will refer to this version of the canopy module as

"ISD" for "Intercepted Snow Densification", while the original version of the canopy module will be referred to as the "Initial Module" or "IM".

In ISD, the density of the intercepted snow ($\rho_{s,int}$; in kg m$^{-3}$) can be estimated as a function of its age ($a_{s,int}$; in days) based on Eq. 16 from Koch et al. (2019):

$$\rho_{s,int} = \rho_{fr} + (\rho_{max} - \rho_{fr})\left(1 - e^{-\frac{a_{s,int}}{\tau}}\right), \tag{1}$$

where $\rho_{fr}$ and $\rho_{max}$ are the fresh snow density and the maximum density for intercepted snow (in kg m$^{-3}$), respectively, and $\tau$ is the shape parameter of the exponential function. We chose to use the model from Koch et al. (2019) because it could to be calibrated using the field data we have available.

Using snow pit density measurements, we determined $\rho_{fr}$ as an average of 11 density measurements of precipitation particles (PP) layers taken during the study period at both sites, which resulted in a value of 80.3 kg m$^{-3}$. For $\rho_{max}$, we used the average

of 69 measurements of RG snow layers during the same period, which led to a value of 280.5 kg m$^{-3}$. $\tau$ was defined such that



$\rho_{s,int}$ is within a 1% deviation of $\rho_{max}$ after $a_{s,int}$ = 30 days, as in Koch et al. (2019). If new snow is intercepted during a time step, $a_{s,int}$ decreases proportionally to the amount of new snow added to the canopy, as follows:

$$a_{s,int} = a_{s,int}\left(1 - \frac{\Delta I\,\Delta t}{(I + \Delta I\,\Delta t)}\right), \tag{2}$$

where $I$ is the interception storage (in mm), $\Delta I$ is the interception rate in mm day$^{-1}$ and $\Delta t$ is the 15 min computational time step (0.010417 day). As long as $\rho_{s,int}$ is less than a threshold density ($\rho_{th}$), the snow stored in the canopy is considered "fresh snow". This means that the microstructure parameters (dendricity ($dd$), sphericity ($sp$), grain diameter ($d_g$) and bond size ($d_b$)) are those of fresh snow according to the parameterization of Lehning et al. (2002b). Similar to before, we empirically set the $\rho_{th}$ value to 152 kg m$^{-3}$, corresponding to the average of 26 density measurements in layers of Decomposed and Fragmented precipitation particles (DF) taken during the study period at both sites. This threshold value is reached after 3.12 days according

to Eq. 1 if no additional snow is intercepted in the meantime. When $\rho_{s,int}$ exceeds the threshold, snow stored in the canopy is considered as RG with $dd$ = 0, $sp$ = 1, $d_g$ = 0.2 mm, and $d_b$ = $d_g$/3 = 0.07 mm. Parameters $dd$, $sp$ and $d_b$ are all default values for RG in SNOWPACK (Lehning et al., 2002b). Grain diameter $d_g$ was set to 0.2 mm, which corresponds to a specific surface area ($SSA$) of 32.7 m$^2$ kg$^{-1}$ for an optical equivalent grain size (Grenfell and Warren, 1999):

$$SSA = \frac{6}{\rho_{ice}d_g}, \tag{3}$$

where $\rho_{ice}$ is the density of ice (917 kg m$^{-3}$). These values of $d_g$ and $SSA$ are within the range of multiple sub–canopy measurements of RG from previous studies (Molotch et al., 2016; Bouchard et al., 2022; Bouchard et al., 2023).

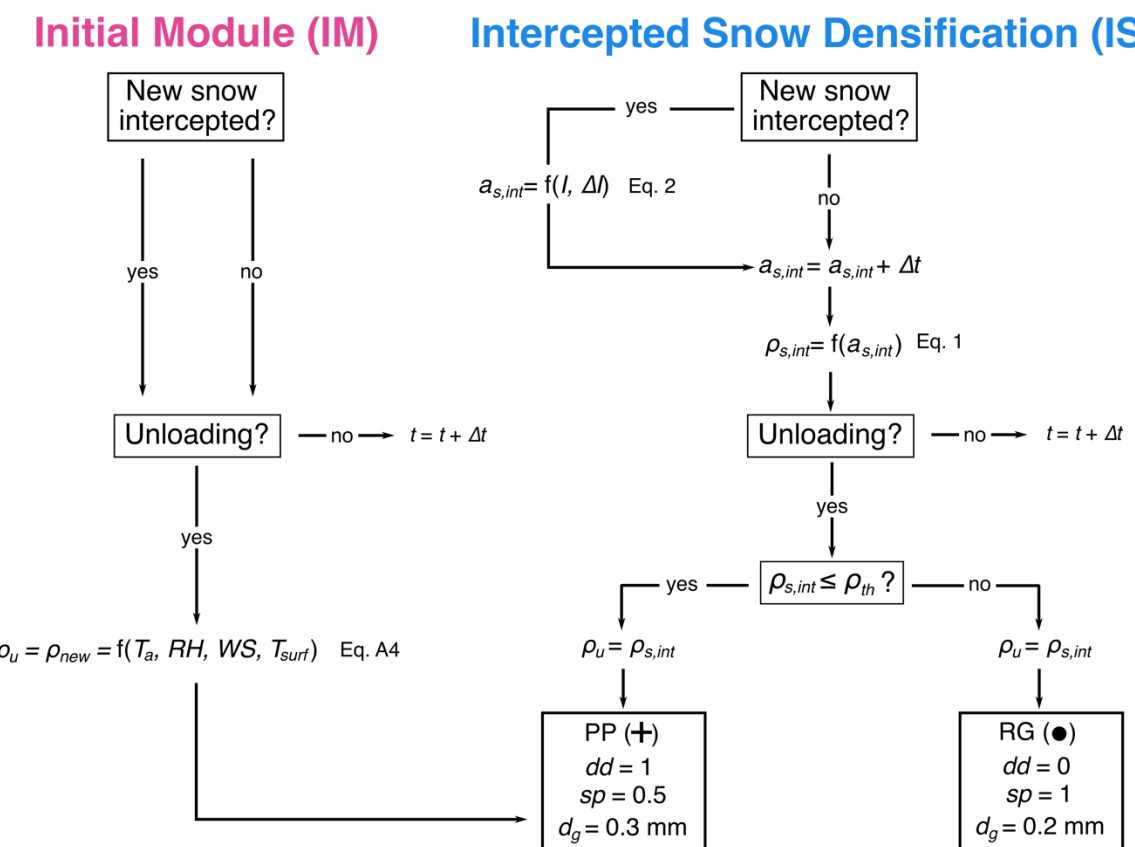

**Figure 2: Schematic description of the Initial canopy Module (IM) and the Intercepted Snow Densification (ISD) with respect to canopy snow properties. With IM, the density of unloaded snow ($\rho_u$) is equivalent to the density of new snow ($\rho_{new}$) and is a function of air temperature ($T_a$), relative humidity ($RH$), wind speed ($WS$) and snow surface temperature ($T_{surf}$). ISD tracks the age of the intercepted snow ($a_{s,\,int}$). When new snow is intercepted, $a_{s,\,int}$ is adjusted as a function of interception storage ($I$) and rate ($\Delta I$) before getting older of one time step ($\Delta t$). Then, the density of the intercepted snow ($\rho_{s,int}$) is computed based on $a_{s,\,int}$. Snow unloads as precipitation particles (PP) snow with the corresponding dendricity ($dd$), sphericity ($sp$), grain diameter ($d_g$) and bond diameter ($d_b$) when $\rho_{s,int}$ is below a threshold density ($\rho_{th}$). If $\rho_{s,int}$ is larger than $\rho_{th}$, snow unloads as rounded grains (RG) with the corresponding $dd$, $sp$, $d_g$ and $d_b$. In both cases, $\rho_u = \rho_{s,int}$. Symbols of PP and RG and taken from the International Classification for Seasonal Snow on Ground (Fierz et al., 2009).**

### 3.2.2 Canopy module parameters

Table 2 shows the canopy parameters that we used for simulations at both sites. Canopy height and LAI were based on field measurements described in Sect. 2.1. The stand basal area ($m^2\ m^{-2}$), which refers to the fraction of total surface area occupied by trunks, was taken from Hadiwijaya et al. (2020) at the MF site. We used half of this value at the BRV site due to the sparser canopy. Since our focus is on snow accumulating below the canopy, we set the direct throughfall fraction to 0 in our simulations at both sites. For the interception capacity factor, parameter $i_{max}$, we applied the value suggested by Schmidt and Gluns (1991) for spruce at both sites. We used the wet, dry and snow–covered canopy albedo and the LAI fraction for the needle layer from





Gouttevin et al. (2015), who showed that these parameters were easily transferable between sites. A full description of all the
canopy parameters can be found in Gouttevin et al. (2015).

| For | MF | BRV | Justification |
|---|---|---|---|
| Canopy height (m) | 9.2 | 15 | measurements |
| LAI ($m^2$ $m^{-2}$) | 4.8 | 1.6 | |
| Stand basal area ($m^2$ $m^{-2}$) | 0.005 | 0.0025 | Hadiwijaya et al. (2020) |
| Direct throughfall fraction (–) | 0 | 0 | sub–canopy simulations |
| $i_{max}$ (mm $m^{-2}$) | 5.9 | 5.9 | Schmidt and Gluns (1991) |
| Dry and wet canopy albedo (–) | 0.11 | 0.11 | Gouttevin et al. (2015) |
| Snow cover canopy albedo (–) | 0.35 | 0.35 | |
| Needle layer LAI fraction (–) | 0.5 | 0.5 | |

**Table 2: User–provided model parameters for the canopy module**

### 3.3 Soil parameterization

In SNOWPACK, snow and soil layers form one continuous column. Snowpack runoff is defined as the liquid water transferred
from the lowermost snow layer to the uppermost soil layer and is computed by applying Darcy's law at the soil–snow interface
(see Eq. 15 from Wever et al. (2014)). The water content in the soil is further calculated based on the van Genuchten model
(see Eq. 3 in Wever et al. (2014)).

The soil was parameterized from in situ measurements (see. Sect. 2.3). The temperature at the bottom of the lowest soil layer,
at a depth of 200 and 182 cm from the surface for MF and BRV simulations, respectively, was defined as a Dirichlet boundary
condition, where the specified temperature corresponds to the deepest measurement. Each soil layer was set with a thickness
of 1 cm, and the properties of lowest observed soil layer were replicated down to the lower boundary at both sites in the
simulation setup. The initial soil parameters provided to SNOWPACK are shown in the Supplementary Material (Tables S1
and S2).

After initializing the soil, we performed a 10 year and 12 year spinup at MF and BRV, respectively, to stabilize the water
content and the thermal regime of the soil (Rodell et al., 2005). In the spinup simulations and in the absence of longer
measurement time series, we looped winter conditions twice from 2018 to 2023 at MF and three times from 2019 to 2023 at
BRV to obtain a state of equilibrium for the soil.

### 3.4 Forcing data

In our study, SNOWPACK is driven in offline mode at the point scale (1–D) using local meteorological observations of $T_a$,
$RH$, $WS$, $SWR{\downarrow}$, $LWR{\downarrow}$, $P$, and precipitation phase ($P_{ph}$) data, and run at a 30 min time step. In the Bernard River Valley, we
used the hourly precipitation from the ERA5-land reanalysis (Muñoz-Sabater et al., 2021), which we equally divided into 30
min time steps. The ERA5-Land reanalysis shows a good agreement with a Hydro-Quebec gauge located 20 km east of the
study site for the annual cumulative precipitation (Fig. S3). The Hydro-Quebec gauge was not used as forcing data because it
only measured precipitation at the mm resolution. A linear transition with dual temperature thresholds at 0 and 2°C was first
used to define the phase of precipitation. Then, the phase was set as rain or snow when precipitation occurred, based on time–



lapse images. Note that in the model, fluxes were calculated perpendicular to the slope accordingly to the radiation measurements above the canopy.

### 3.5 Model evaluation

We first evaluated the model performance in simulating snow height using the root–mean–square error (RMSE; cm) and the bias relative to the observations ($p_{bias}$; %). We also evaluated the difference in snow disappearance date ($\Delta$SDD; days). We

further calculated the RMSE of $T_{surf}$ (in °C) at MF in 2018–19 and from 2020 to 2023. Validating simulated $T_{surf}$ provides an estimate of how well the overall energy balance of the snowpack is captured. Then, we compared the simulated and observed vertical profiles of density and grain type and evaluated the agreement between both on a score from 0 to 1 (agreement score) based on the method described by Lehning et al. (2001). Finally, we visually compared the simulated stratigraphy with field observations of melt–freeze and ice layers as a semi–quantitative analysis of the number of correctly simulated melt–freeze

layers.

| Parameter | low | assigned | high |
|---|---|---|---|
| $\rho_{fr}$ (kg m$^{-3}$) | 30 | 80.3 | 140 |
| $\rho_{max}$ (kg m$^{-3}$) | 200 | 280.5 | 350 |
| $\rho_{th}$ (kg m$^{-3}$) | 100 | 152 | 200 |
| $d_g$ (mm) | 0.1 | 0.2 | 0.4 |

**Table 3: Values of canopy snow parameters used for the sensitivity analysis**

### 4 Results

### 4.1 Climatic conditions

### 4.1.1 Snow height and ROS events

At both sites, snow began to accumulate between late October and mid–November, and started to decline between late March and early April (Fig. 3). In general, the snowpack disappeared in late May at MF and in early May at BRV, for a snow cover duration on average 15 days longer at the MF site (197 vs 182 days). Also, more snow tended to accumulate at the MF site than at the BRV site. However, winter 2020–21 at MF was by far the year with the lowest snow accumulation among all site–years. This winter was an exceptionally warm and low–snow year at MF, and is analyzed in detail in Bouchard et al. (2023).



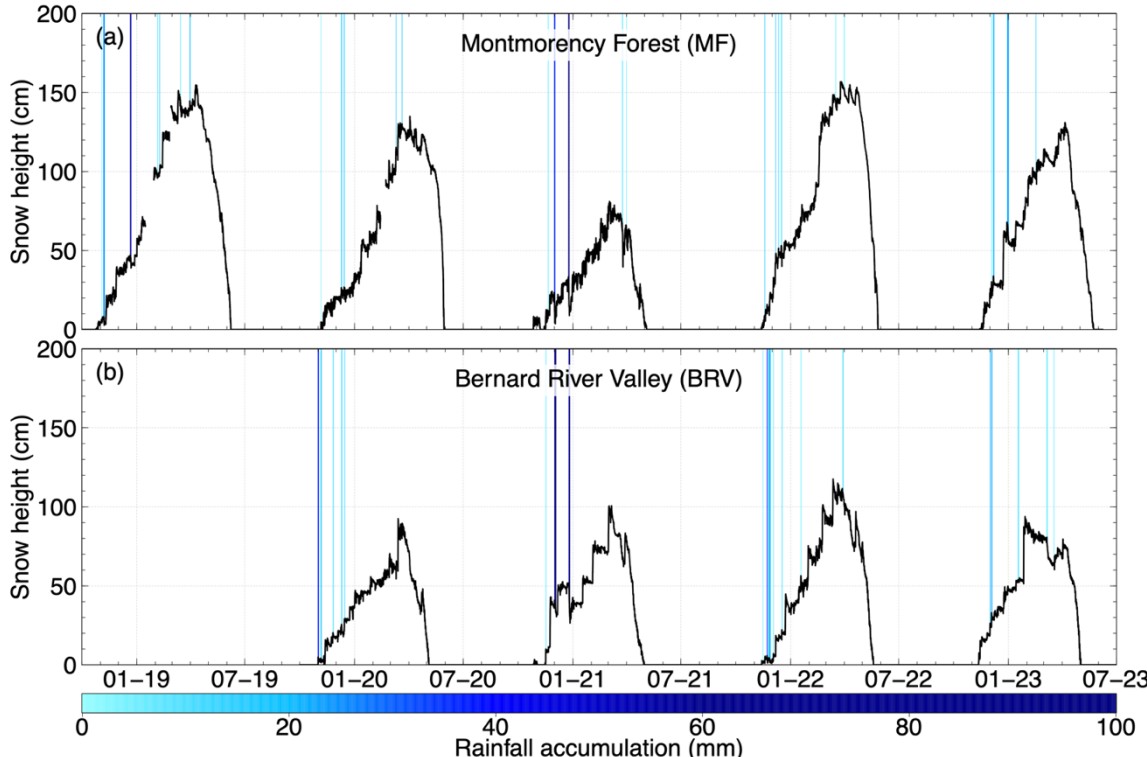

**Figure 3: Observed snow heights from October 2018 to July 2023 at Montmorency Forest (MF; a) and from November 2019 to July 2023 at Bernard River Valley (BRV; b). Vertical blue bars show the rain-on-snow (ROS) events recorded from November to March at both sites (47 in total), with darker blue indicating greater rainfall accumulation per event (see color code). The precipitation phase was determined using time–lapse cameras.**

We observed 27 ROS events at MF and 20 at BRV from November to March throughout the study period (Figs. 3 and 4). 22 ROS events were less than 10 mm, and only 7 of them exceeded 30 mm. We recorded one event of more than 100 mm at each site, on 24 December 2020 at MF (106 mm) and on 1 December 2020 at BRV (113 mm). In total, 31 ROS events occurred from October to December and 16 from January to March. Few events were observed when the air temperature was below freezing, none of them being larger than 25 mm. In general, ROS events occurred when the air temperature was between 0°C and 5°C, although a few large events took place when the temperature was higher than 5°C. Long ROS events did not lead necessarily to important rainfall accumulation, as shown by the large dots in the lower part of Fig. 4. In contrast, events of more than 30 mm all lasted longer than 12 hours.





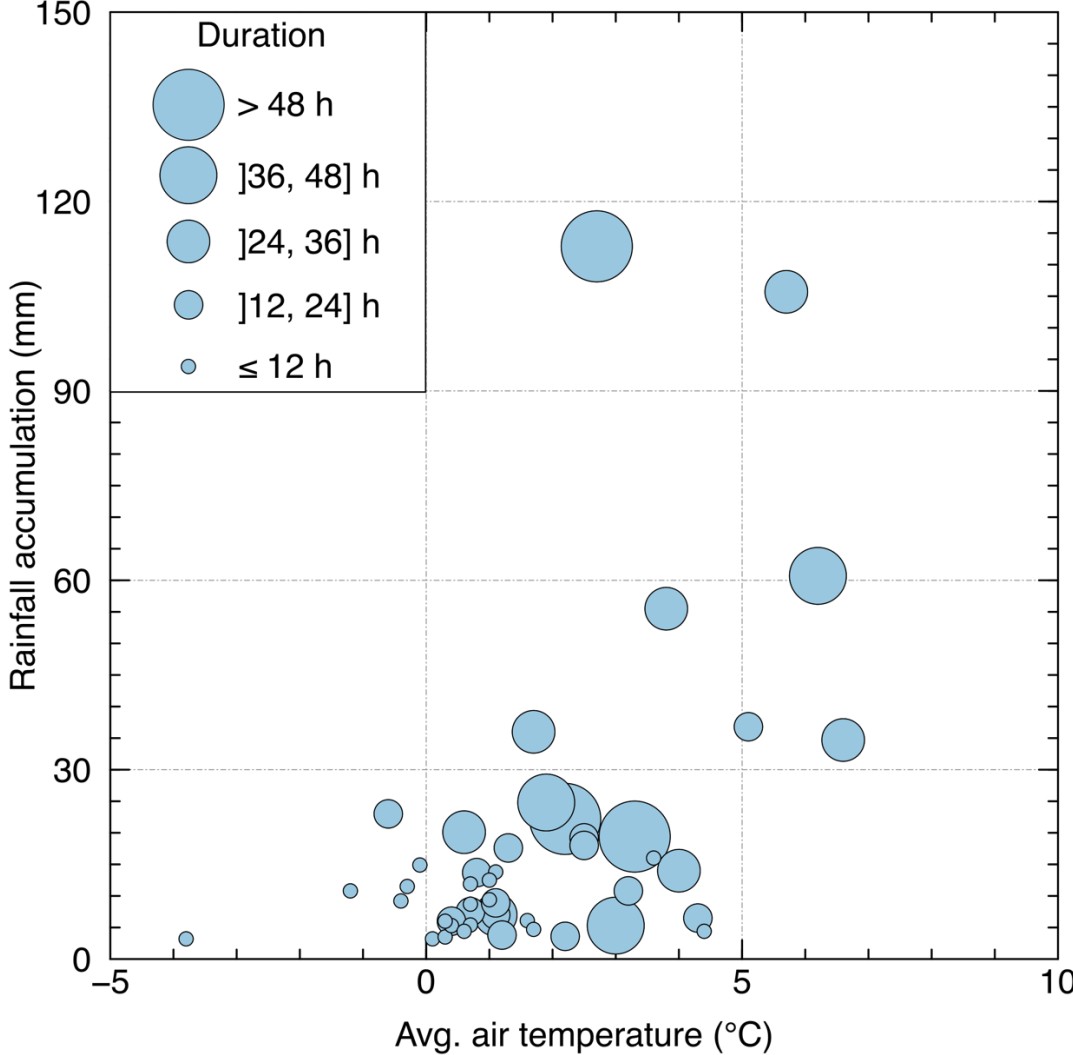

**Figure 4: Total liquid water accumulation, mean air temperature, and duration of each ROS event observed for November to March**
**period from 2018 to 2023 in the Montmorency Forest (MF) and from 2019 to 2023 in the Bernard River Valley (BRV).**

**4.1.2 Snowpack observations**

During the study period, 18 melt–freeze layers were identified at MF and 14 at BRV (Table 4). Melt–freeze layers formed at
the surface by diurnal melt–freeze cycles are excluded from the list, as are layers formed by ROS events from April to June.
Also, since the observations were made under the forest cover, it is unlikely that shortwave radiative melting followed by
nocturnal longwave radiative cooling led to a melt–freeze layer (Höller, 2001; Malle et al., 2019). Therefore, it is assumed that
the melt–freeze layers listed in Table 4 were all formed at the snow surface and originated from a ROS event between
November and March.



| Site | Layer ID | Formation date | Avg. height (cm) | Avg. thickness (cm) | N. obs |
|------|----------|----------------|------------------|---------------------|--------|
| **MF** | MF18–19a+ | 6 Nov. 2018 | 0 | 7.7 | 26 |
| | MF18–19b | 21 Dec. 2018 | 40 | 5.2 | 21 |
| | MF18–19c | 5 Feb. 2019 | 85.2 | 8.9 | 14 |
| | MF18–19d | 15 Mar. 2019 | 140.3 | 5.3 | 6 |
| | MF19–20a+ | 5 Nov. 2019 | 0 | 3.3 | 3 |
| | MF19–20b | 9 Dec. 2019 | 21 | 3.5 | 2 |
| | MF20–21a | 30 Nov. 2020 | 9 | 2 | 1 |
| | MF20–21b+ | 24 Dec. 2020 | 0 | 6.5 | 2 |
| | MF20–21c | 24 Mar. 2021 | 43 | 2 | 1 |
| | MF21–22a+ | 18 Nov. 2021 | 0 | 6 | 4 |
| | MF21–22b | 6 Dec. 2021 | 25.3 | 1.7 | 3 |
| | MF21–22c | 11 Dec. 2021 | 32.5 | 4.5 | 4 |
| | MF21–22d | 17 Mar. 2022 | 114 | 4 | 1 |
| | MF21–22e | 31 Mar. 2022 | 133 | 2 | 1 |
| | MF22–23a | 3 Dec. 2022 | 18.3 | 7 | 3 |
| | MF22–23b | 6 Dec. 2022 | 31.7 | 3 | 3 |
| | MF22–23c | 30 Dec. 2022 | 51.7 | 2.7 | 3 |
| | MF22–23d | 15 Feb. 2023 | 99.3 | 1.6 | 3 |
| **BRV** | BRV19–20a+ | 31 Oct. 2019 | 0 | 3 | 1 |
| | BRV19–20b | 25 Nov. 2019 | 29 | 1 | 1 |
| | BRV20–21a+ | 16 Nov. 2020 | 0 | 7.5 | 2 |
| | BRV20–21b | 1 Dec. 2020 | 24.5 | 4 | 2 |
| | BRV20–21c | 25 Dec. 2020 | 38.5 | 2.5 | 2 |
| | BRV21–22a+ | 22 Nov. 2021 | 0 | 5.5 | 2 |
| | BRV21–22b | 3 Dec. 2021 | 8 | 4 | 1 |
| | BRV21–22c | 17 Dec. 2021 | 23.5 | 3 | 2 |
| | BRV21–22d | 18 Jan. 2022 | 54 | 4 | 2 |
| | BRV21–22e | 29 Mar. 2022 | 104 | 4 | 1 |
| | BRV22–23a | 1 Dec. 2022 | 25 | 3 | 1 |
| | BRV22–23b | 17 Jan. 2023 | 35 | 1 | 1 |
| | BRV22–23c | 6 Mar. 2023 | 42 | 4 | 1 |
| | BRV22–23d | 18 Mar. 2023 | 57 | 4 | 1 |

\*The formation date is assumed to be the first rain-on-snow event since the previous layer was observed for the first time.
**Table 4: All observed melt–freeze layers at Montmorency Forest (MF; 18) and at Bernard River Valley (BRV; 14) that are assumed to have formed on the snow surface after a rain-on-snow (ROS) event. Details include the formation date\*, the average height of the bottom of the layer, its average thickness, and the number of times each layer was observed during snow pit experiments. The cross indicates a basal melt–freeze layer.**

At both sites, melt–freeze layers were always observed at the base of the snowpack, except for winter 2022–2023. Not surprisingly, most layers were formed from October to December, as ROS events were more frequent during that period. Therefore, more melt–freeze layers were observed in the bottom 50 cm of snow. Note that the average thickness of the melt–freeze layers was highly variable, ranging from 1 to 9 cm.

Several small frozen percolation channels between two layers (BRV22–23c and BRV22–23d from Table 4) are shown in Fig. 5a–b, indicating that preferential flow occurred through multiple small channels within a short distance. In Fig. 5c, three small percolation channels were observed above a melt–freeze layer (MF21–22b from Table 4), which was itself above one larger percolation channel. This suggests that less preferential flow paths form deeper in the snowpack where snow grains are coarser, consistent with the observations of Katsushima et al. (2013). Pockets of snow with liquid water were also observed at various depths in the snowpack (Fig. 5d). The absence of a continuous vertical wetting between these pockets suggests that water was





transported downward by preferential flow. Figures 5e–f show ice clumps on the surface of the snowpack during snowmelt. These ice structures appear to be residual preferential channels melting at slower rates than the surrounding snow, as also reported by Teich et al. (2019).

**Figure 5: (a) Melt–freeze layers and percolation channels (30 March 2023 – BRV). (b) Closer view of the leftmost percolation channel observed in (a). (c) Preferential flow paths are less numerous but larger in diameter deeper in the snowpack where the snow grains are coarser. (10 March 2022 – MF). (d) Pockets of liquid water within the snowpack (30 March 2022 – BRV). (e–f) Residual percolation channels remaining on top of the sub–canopy snowpack during the snowmelt period (28 April 2022 – MF).**



Figure 6 shows observations of air temperature, snow temperature, and temperature at the soil–snow interface for a ROS event at each site. In the first event (Fig. 6a–c), air temperature rose to 0°C on the night of 15–16 February 2023 at MF, before falling to −23°C over the next two days. A total of 11.5 mm of rain was measured over 11 hours when the air temperature was highest. The top layers of the snowpack reached 0°C quickly after the start of the ROS event. The snow underneath gradually warmed

from above, but never reached 0°C, except for the temperature probe at 15 cm height, which reached 0°C some 24 hours later. This vertically inhomogeneous warming pattern suggests preferential flow. A constant soil–snow temperature around −0.3°C during the ROS suggests that percolation did not reach the ground.

In the second event (Fig. 6d–f), a temperature rise above 0°C at BRV on the 6 and 7 March 2023 triggered a phase change of the precipitation, so that 5.3 mm of rain were recorded during this event that lasted for almost 40 hours. This ROS event

unfolded in two main steps. First, rain and warm air warmed the top snow layers to 0°C, from the surface down to a snow height of 45 cm. Then, additional rain led to a rapid increase of the soil–snow interface temperature to 0°C before snow layers above. This supports the idea that water reached the ground through preferential flow. Note that given the important water percolation in the snowpack from that event, it is likely that ERA5 underestimated rainfall accumulation.







**Figure 6: Evolution of the 2.5 m high air temperature (a and d), snowpack temperature profile (b and e), and soil–snow interface temperature (c and f) during and after the ROS event of 16 February 2023 at Montmorency Forest (MF; a–c) and during the ROS of 6 March 2023 at Bernard River Valley (BRV; d–f). Liquid and solid precipitation are shown in panels (b) and (e) and gray–red areas in the temperature profile indicate a temperature of 0°C. Snowpack temperature is vertically interpolated between measurement heights (every 15 cm) and the observed temperature was forced to a maximum of 0°C. Snow height is shown by the black line on panel (e) since snow surface temperature was not measured at BRV.**



## 4.2 Evaluation of SNOWPACK

SNOWPACK is hereby evaluated with the initial interception module (IM). SNOWPACK with IM overestimates the snow height under the canopy at MF with a RMSE of 17.3 cm and a $p_{bias}$ of 19.5% (Fig. 7). Also, the simulated snowpack melts out later than the observations ($\Delta$SDD = 5.5 days). The model performance is lower at BRV than at MF, which could be partly

explained by the use of ERA5-Land reanalysis as precipitation input to the model versus local high–quality measurements at MF. Overall, scores at MF are deemed acceptable given that snow height is highly heterogeneous spatially under the canopy (Parajuli et al., 2020).

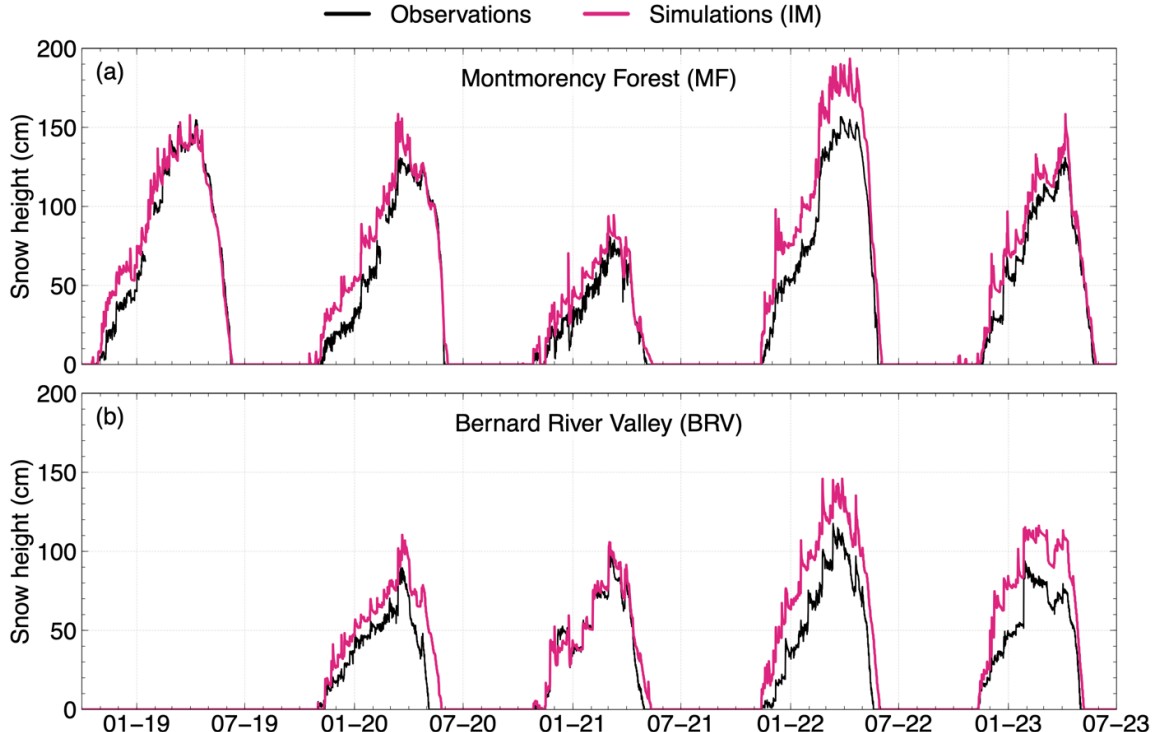

**Figure 7: Comparison of observed (black) versus modeled snow height using SNOWPACK with the Initial canopy Module (IM;**
**pink) in the Montmorency Forest (a) and in the Bernard River Valley (b).**

As shown in Fig. 8a, SNOWPACK with IM simulates $T_{surf}$ with an RMSE of 1°C and a $p_{bias}$ of 0.5% in 2018 and from 2020 to 2023 (winters 2018–19, 2020–21 and 2021–22 are presented in Supplementary Material). However, there is some discrepancy between the model and the observations when looking at a daily time scale. (Fig. 8b). In this example, SNOWPACK accurately estimates $T_{surf}$ from 7 to 10 March, corresponding to cloudy conditions. However, in the following

three days, corresponding to clear sky conditions, the model overestimates $T_{surf}$ during the day and at night. In general, the results suggest that the energy balance is adequately described by SNOWPACK, so we can trust the timing of the onset of melt. For example, the observed rise to 0°C in early April is well reproduced by the model (Fig. 8a). Figure S4 shows that this is also the case for the other years.




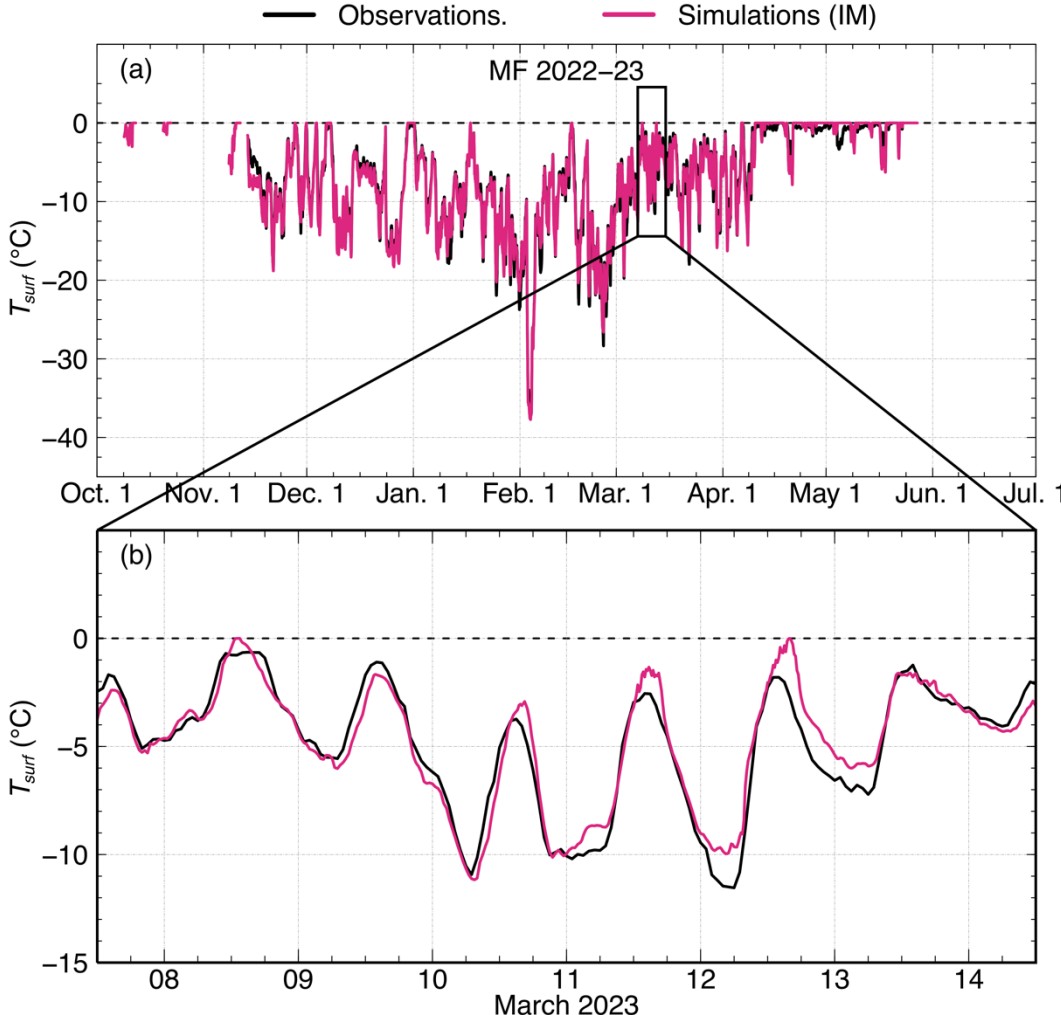

**Figure 8: Comparison of observed (black) versus modeled snow surface temperature using SNOWPACK with the Initial canopy Module (IM; pink) during the winter of 2022–23 (a) at Montmorency Forest (MF). (b) Closer look at the observed and simulated surface temperatures during the week of 7 to 14 March 2023.**

The model shows an average agreement score of 0.79 for snow density profiles compared to observations at MF (Fig. 9a). Density agreement scores are generally constant throughout the season, except during winter 2018–19 at MF, where the performance decreases during the melt period when the density is more uniform vertically. The model performs better at simulating the upper snow layers than the lower layers, where SNOWPACK tends to overestimate the density (Figs. S5 to S11). This is not surprising given that SNOWPACK does not account for upward convective water vapor fluxes (Domine et al., 2019) or vapor diffusion in the version that we used (Jafari et al., 2020). At BRV, the agreement score for snow density is 0.77 and the vertical density profile is also better represented at the top than at the bottom (Figs. S12 to S15). Although we could evaluate the model's performance based on only on six profile observations at BRV (Fig. 9b), these results are similar to those from MF and support the conclusion reached for this site.





SNOWPACK does not simulate grain type as well as snow density, with an average agreement score of 0.59 at MF (Fig. 9c). Lower agreement scores for grain type are also obtained for simulations at BRV (Fig. 9d). As with density measurements, there is no clear difference in performance between seasons for the grain type. In winter 2018–19, there is a strong decrease in agreement score from the beginning of January to the end of April. This decrease in performance occurs immediately after a ROS on 21 December 2018. SNOWPACK simulates the complete wetting of the snow column, whereas a thinner melt–freeze layer was observed (MF18–19b in Table 4). The score then increases during the 2019 snowmelt, as the melt forms observed in the field were also mostly simulated by the model. This is not a surprise given that melt forms are more uniform than other grain types (Colbeck, 1982).

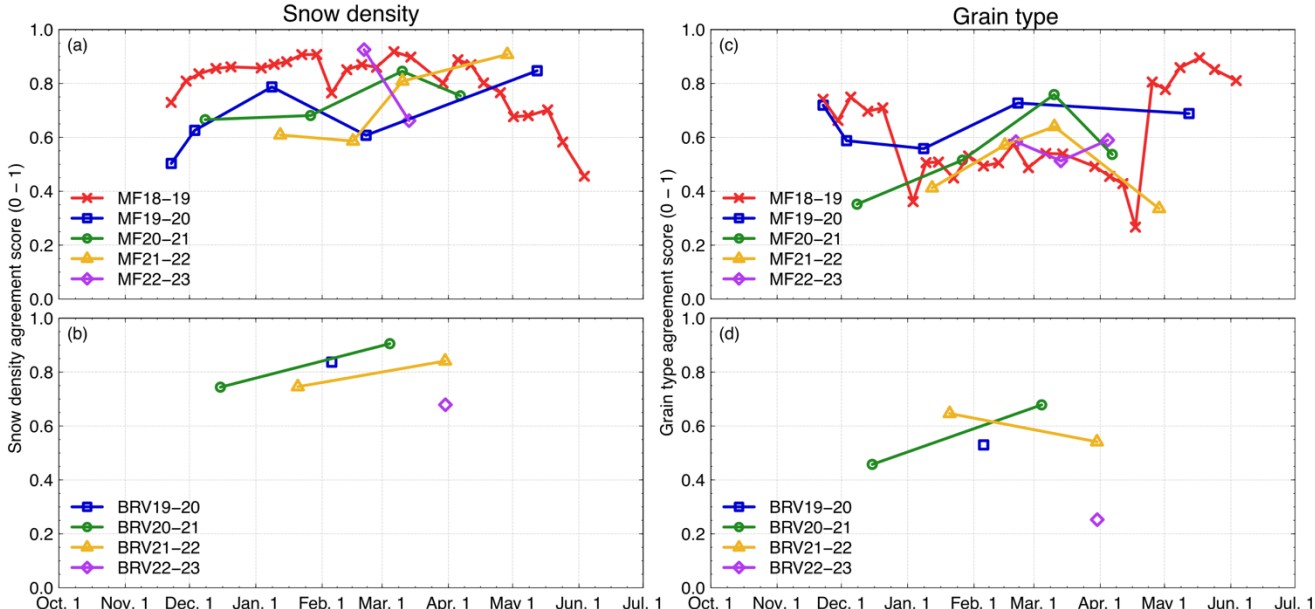

**Figure 9: Agreement scores between the SNOWPACK simulations using the Initial canopy Module (IM) and the observations for the snow density profiles (a) and (b) and snowpack stratigraphy (c) and (d). Agreement scores are computed for all snow pit measurements taken between 2018 and 2023 at Montmorency Forest (MF) and between 2019 and 2023 at Bernard River Valley (BRV).**

## 4.3 Age–based intercepted snow densification

### 4.3.1 Effects on snow unloading

The correspondence between snow in trees inferred from photographs and the variations in modeled snow canopy storage (Fig. 10a) suggests that the removal of intercepted snow by unloading and evaporation is well captured by the model. However, ISD causes the unloaded snow to have a significantly larger density compared to IM (Fig. 10b). Note that the decrease in intercepted snow density is due to the interception of new snow, which has generally a lower density than the snow already stored in the canopy.





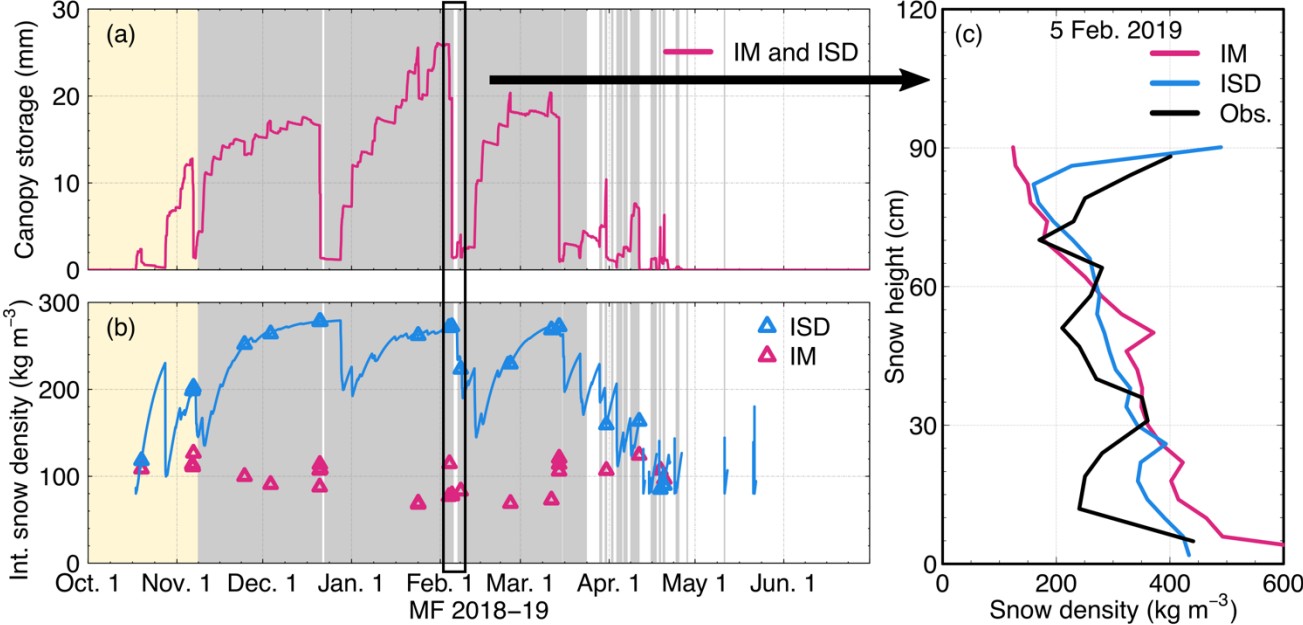

**Figure 10: Canopy storage and intercepted snow density as simulated by SNOWPACK at Montmorency Forest (MF) in 2018–19 with the Initial canopy module (IM; pink) and the Intercepted Snow Densifcation (ISD; blue; a–b). Triangles in (b) show the simulated snow density as it unloads from the canopy. The gray shaded areas in (a) and (b) indicate the presence of snow in the canopy as observed by the time lapse cameras. No photos were taken at the beginning of the season, indicated by the yellow shaded area. (c) Snow density profiles simulated with both versions of SNOWPACK and compared to observations on 5 February 2019. The simulated profiles are aggregated to a 4 cm vertical resolution and stretched to match the observed snow height.**

### 4.3.2 Effects on snowpack properties

As shown in the density profile comparison (Fig. 10c), ISD produces near–surface density values in better agreement with observations than those of IM, which strongly underestimates density. A higher initial density with ISD is also consequently compensated by lower settling rates. Overall, except for layers resulting from snow unloading, IM and ISD simulate similar density profiles (Figs. S5 to S15).

The use of SNOWPACK with ISD has a greater impact on the simulated grain type as shown in Fig. 11. From early December 2018 to mid–April 2019, we observe, from bottom to top, apart from melt–freeze layers, depth hoar, faceted crystals, rounded grains and recent snow. Most importantly, we observe 4 melt–freeze layers resulting from ROS events (MF18–19a–d in Table 4). With IM, only the basal melt–freeze layer (MF18–19a) is well simulated, with the upper melt–freeze layers either too thick (MF18–19b), absent (MF18–19c), or not generated at the correct height (MF18–19d). In contrast, ISD reproduced all observed melt–freeze layers quite well until late March (Fig. 11c). Other thin melt–freeze layers were formed in April, but these were difficult to identify from snow pit observations and were therefore not used to evaluate model performance. Of the 18 melt–freeze layers observed at MF (Table 4), ISD is able to reproduce 17 of them, compared to 10 by IM (Figs. S16 to S19). At BRV, ISD simulated 10 layers instead of 8 by IM, supporting our results from MF (Figs. S20 to S23).



Since the melt–freeze layers are a small fraction of the snow column, this improvement does not translate into a large improvement in the grain type agreement scores (Table 5), which also indicates a limitation inherent to the formulation of the

agreement score. For snow height, $T_{surf}$, and density profiles, both versions of the canopy module also give similar results (Table 5). This shows that the performance gain of ISD for simulating snow stratigraphy is achieved without compromising the performance with respect to other snow properties.



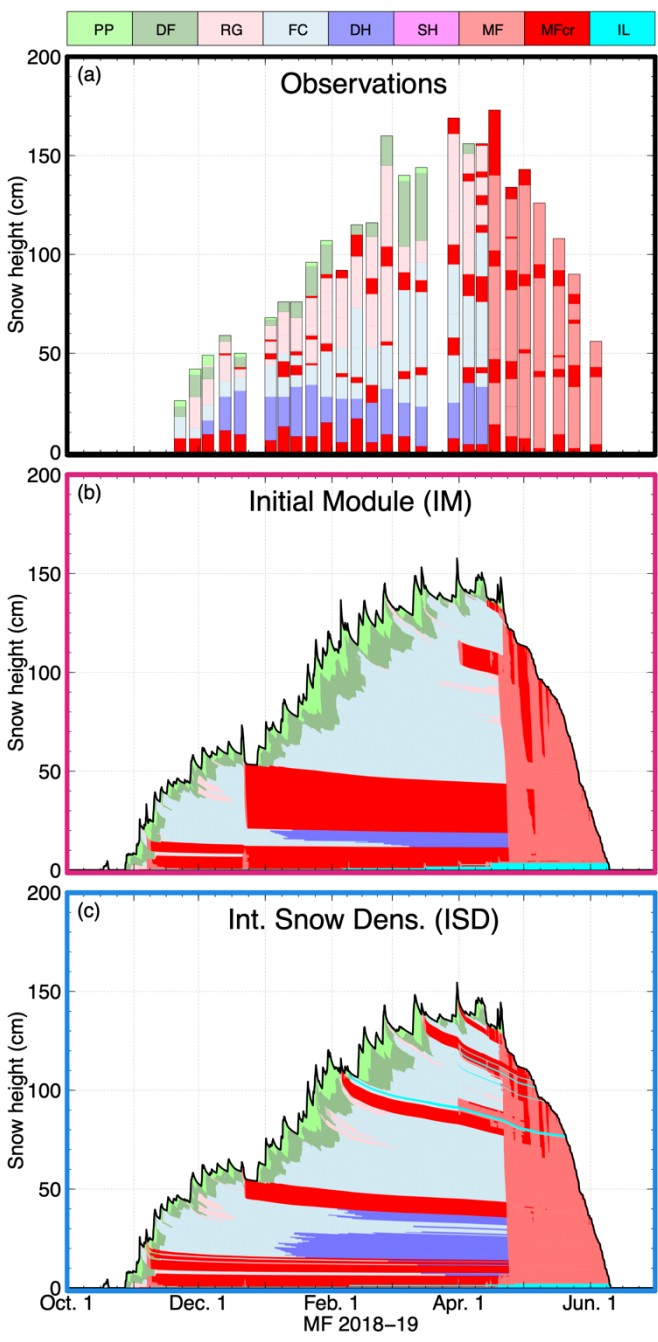

**Figure 11: Sub–canopy snowpack stratigraphy in the Montmorency Forest (MF) during winter 2018–2019 as (a) observed from 26 snow pits and simulated with SNOWPACK using (b) the Initial canopy Module (IM) and (c) the Intercepted Snow Densification module (ISD). The following grain types are present: PP (precipitation particles), DF (decomposed and fragmented PP), RG (rounded grains), FC (faceted crystals), DH (depth hoar), SH (surface hoar), MF (melt forms), MFcr (melt–freeze layers) and IL (ice layers). MFcr and IL are shown in darker red and cyan, respectively. The color code for snow grain type is taken from the International Classification for Seasonal Snow on the Ground (Fierz et al., 2009).**




| Model version | Site | Variable | RMSE (cm;°C)* | $p_{bias}$ (%) | ΔSDD (days) | Agr. Score (0–1) | N. layers detected** |
|---|---|---|---|---|---|---|---|
| IM | MF | Snow height | 17.3 | 19.5 | +5.5 | - | - |
| | | $T_{surf}$ | 1.0 | 0.5 | - | - | - |
| | | Dens. prof. | - | - | - | 0.79 | - |
| | | Grain type | - | - | - | 0.59 | - |
| | | Melt−freeze layers | - | - | - | - | 10 |
| | BRV | Snow height | 22.5 | 36.0 | +11.9 | - | - |
| | | Dens. prof. | - | - | - | 0.77 | - |
| | | Grain type | - | - | - | 0.52 | - |
| | | Melt−freeze layers | - | - | - | - | 8 |
| ISD | MF | Snow height | 17.9 | 20.9 | +5.3 | - | - |
| | | $T_{surf}$ | 1.0 | 0.5 | - | - | - |
| | | Dens. prof. | - | - | - | 0.80 | - |
| | | Grain type | - | - | - | 0.62 | - |
| | | Melt−freeze layers | - | - | - | - | 17 |
| | BRV | Snow height | 22.0 | 35.5 | +11.5 | - | - |
| | | Dens. prof. | - | - | - | 0.78 | - |
| | | Grain type | - | - | - | 0.51 | - |
| | | Melt−freeze layers | - | - | - | - | 10 |

*RMSE in cm for snow height and in °C for the snow surface temperature
**out of 18 layers at MF and 14 at BRV
**Table 5: Summary of the performance of SNOWPACK for simulating the sub–canopy snowpack in Montmorency Forest (MF) and in the Bernard River Valley (BRV) for both versions of the canopy module (IM and ISD). The summary includes the evaluation of snow height and snow surface temperature ($T_{surf}$; RMSE, $p_{bias}$), density and grain type profiles (agreement score) and the number**
**of ROS–induced melt–freeze layers detected. All metrics cover the entire study at both sites, except $T_{surf}$ which excludes winter 2019–20.**

### 4.3.3 Case of the ROS events on 5 and 8 February 2019

To assess the timing of the effects of intercepted snow densification in simulations, we carefully examine two ROS events that took place on 5 and 8 February 2019 at the MF site (see Fig. 12). The snow that unloaded at the beginning of both ROS events
produced a thick, low–density snow layer when simulated with IM (Fig. 12a). This resulted in liquid water that quickly percolated through the snowpack by preferential flow (Fig. 12b). Since water remained in the preferential flow domain, the snow column temperature stayed below 0°C and no melt–freeze layer was created (Fig. 12c–d). As preferential flow conveys water downward rapidly, the model generated runoff a few hours only after the onset of rainfall for both ROS (Fig. 12e). However, runoff volumes were smaller than rainwater volumes, indicating that some of precipitation was retained in the
snowpack.

With the ISD module, unloaded snow triggered by rainfall created a dense layer of snow on top of the snowpack (Fig. 12f), creating fine–over–coarse conditions that restricted downward water flow. This caused water to accumulate in the preferential flow domain (Fig. 12g) until a small fraction of it percolated through when the pressure head exceeded the water entry pressure of the underlying layer with increasing grain size. The other fraction of water was transferred to the matrix flow domain when
$\theta_{TH}$ was exceeded (Fig. 12h), which led to the formation of a melt–freeze layer (Fig. 12i). This is in line with the observed



stratigraphy from 5 February 2019 presented on the right of Fig. 12i. Since water was mostly retained in the snowpack to further freeze, the ROS events resulted in almost no runoff (Fig. 12j).

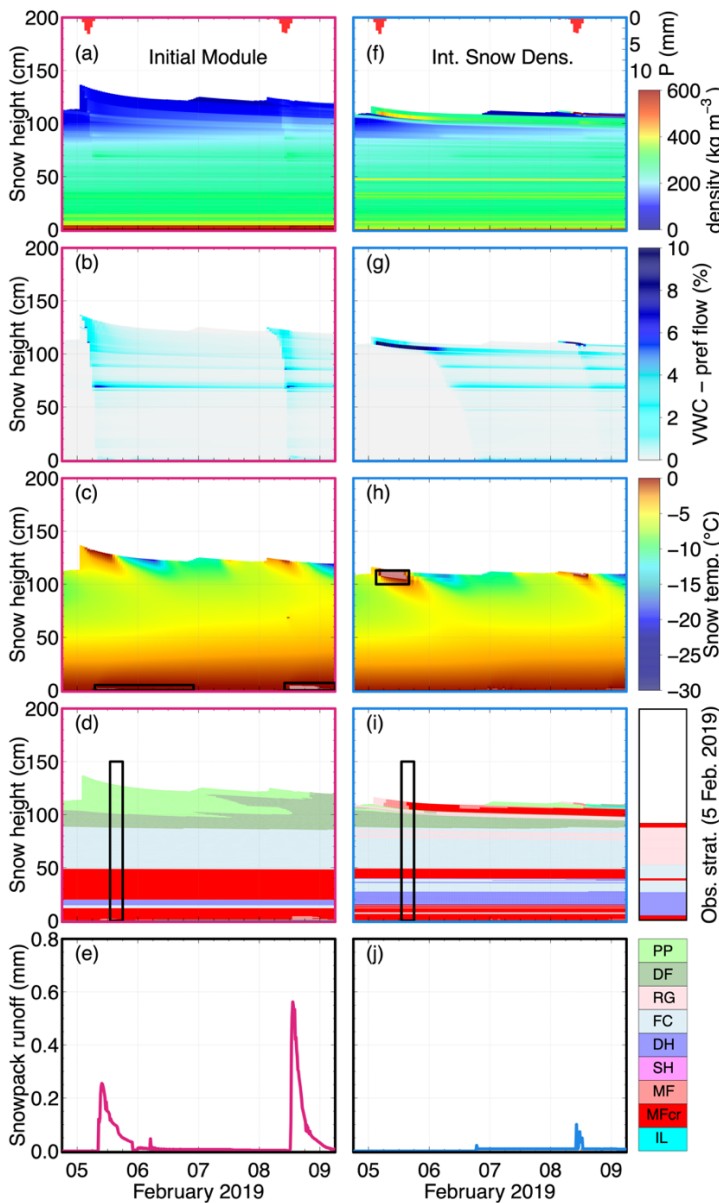

**Figure 12: The 5 February 2019 rain-on-snow (ROS) event in the Montmorency Forest (MF) simulated with SNOWPACK using the Initial canopy Module (IM; left column) and the Intercepted Snow Densification module (ISD; right column). Panels (a) and (f) show liquid precipitation (P) on top and the evolution of snow density profile during the ROS. Panels (b) and (g) show the evolution of volumetric liquid water content (VWC) in the preferential flow domain. Panels (c) and (h) present the evolution of snow temperature and the presence of VWC in the matrix flow domain in shaded white (also highlighted by black rectangles). Panels (d) and (i) show the snowpack stratigraphy during the ROS, along with grain type observations from 5 February 2019. The black rectangles mark the simulated profile on this date. Finally, panels (e) and (j) show the snowpack runoff resulting from the ROS events.**





### 4.3.4 Effects on snowpack runoff

The version of the canopy module used (IM vs ISD) dictates the simulated runoff resulting from ROS events from November

through March (Fig. 13a). Compared to the simulations with IM, 70% of the events simulated with ISD were smaller and

generated 27% less runoff volume. Of the 47 events, only 12 with IM and 9 with ISD have runoff greater than rainfall

accumulation. This means that most ROS events during the accumulation period result in rainwater freezing in the snowpack

rather than ROS events triggering snowmelt. For two major rainfall events at MF, the runoff simulated by ISD was 54 mm (21

December 2018) and 44 mm (24 December 2020), less than that simulated by IM. In both cases, IM generated snowmelt, while

ISD caused rainwater to freeze in the snowpack, demonstrating that the canopy module has an impact on the simulated runoff.

Most importantly, it suggests that the processes in the canopy can strongly influence runoff generation during rain-on-snow

events, which can be hard to quantify or simulate in a model. The caveat here is that the runoff cannot be validated.

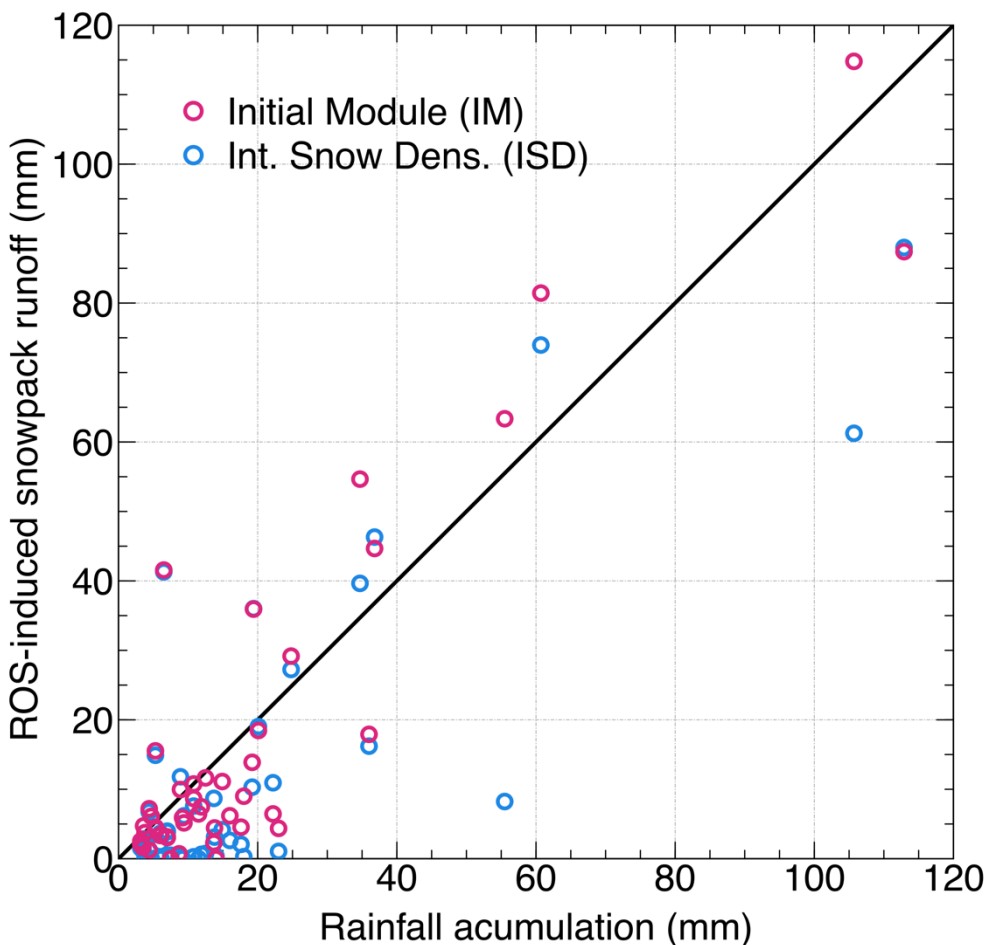

**Figure 13: Simulated runoff from the Initial canopy Module (IM) and the Intercepted Snow Densification module (ISD) for all 47 rain-on-snow events from November to March at both sites. The black line shows the 1:1 relationship.**





## 4.4 Sensitivity analysis


Of the 26 snow pits of winter 2018–19 at MF, melt–freeze layers were reported 81 times by the observer (Table 6). The number of simulated melt–freeze layers is not very sensitive to canopy snow parameters with hits ranging from 57 ($\rho_{max}$ = 200 kg m$^{-3}$) to 65 ($\rho_{th}$ = 100 kg m$^{-3}$). However, the number melt–freeze layers simulated with the correct thickness is more sensitive, with hits ranging from 21 ($d_g$ = 0.1 mm) to 39 ($\rho_{th}$ = 200 kg m$^{-3}$). Compared to IM, ISD detects more layers and simulates the

thickness better, regardless of the sensitivity analysis parameters that we used. This suggests that ISD is more capable of simulating melt–freeze layers with the correct thickness than IM.

As shown in Fig. 14, the model sensitivity is more important for the runoff simulations than for the melt–freeze layers. Except for one point corresponding to an event in late March, both $\rho_{fr}$ and $\rho_{th}$ have a very small effect on the average cumulative runoff from a ROS event (± 0.3 mm). In contrast, simulations performed with a reduced and increased $\rho_{max}$ result in event–

based cumulative runoff that is on average 2.5 mm lower and 2.1 mm higher, respectively, than for simulations with all parameters unchanged. Similarly, decreasing and increasing $d_g$ result in simulated cumulative runoff that is 2.9 mm higher and 4.6 mm lower, respectively, on average. This indicates that the average grain size when snow unloads has a greater effect on ROS–induced runoff than the intercepted snow density.

| Simulations | Hits | Hits with correct thickness |
|---|---|---|
| IM | 43 | 17 |
| ISD (default par.) | 61 | 34 |
| ISD ($\rho_{fr}$= 30 kg m$^{-3}$) | 59 | 35 |
| ISD ($\rho_{fr}$= 140 kg m$^{-3}$) | 64 | 33 |
| ISD ($\rho_{max}$= 200 kg m$^{-3}$) | 57 | 27 |
| ISD ($\rho_{max}$= 350 kg m$^{-3}$) | 62 | 34 |
| ISD ($\rho_{th}$= 100 kg m$^{-3}$) | 65 | 37 |
| ISD ($\rho_{th}$= 200 kg m$^{-3}$) | 63 | 39 |
| ISD ($d_g$ = 0.1 mm) | 63 | 21 |
| ISD ($d_g$ = 0.4 mm) | 60 | 33 |

**Table 6: Number of observed melt–freeze layers reproduced ("hits") and simulated with the right thickness by SNOWPACK with**
**IM, ISD and ISD with the modified canopy parameters identified (see Table 2) in winter 2018–19 at MF. Note that 81 melt–freeze layers were observed in snow pits.**





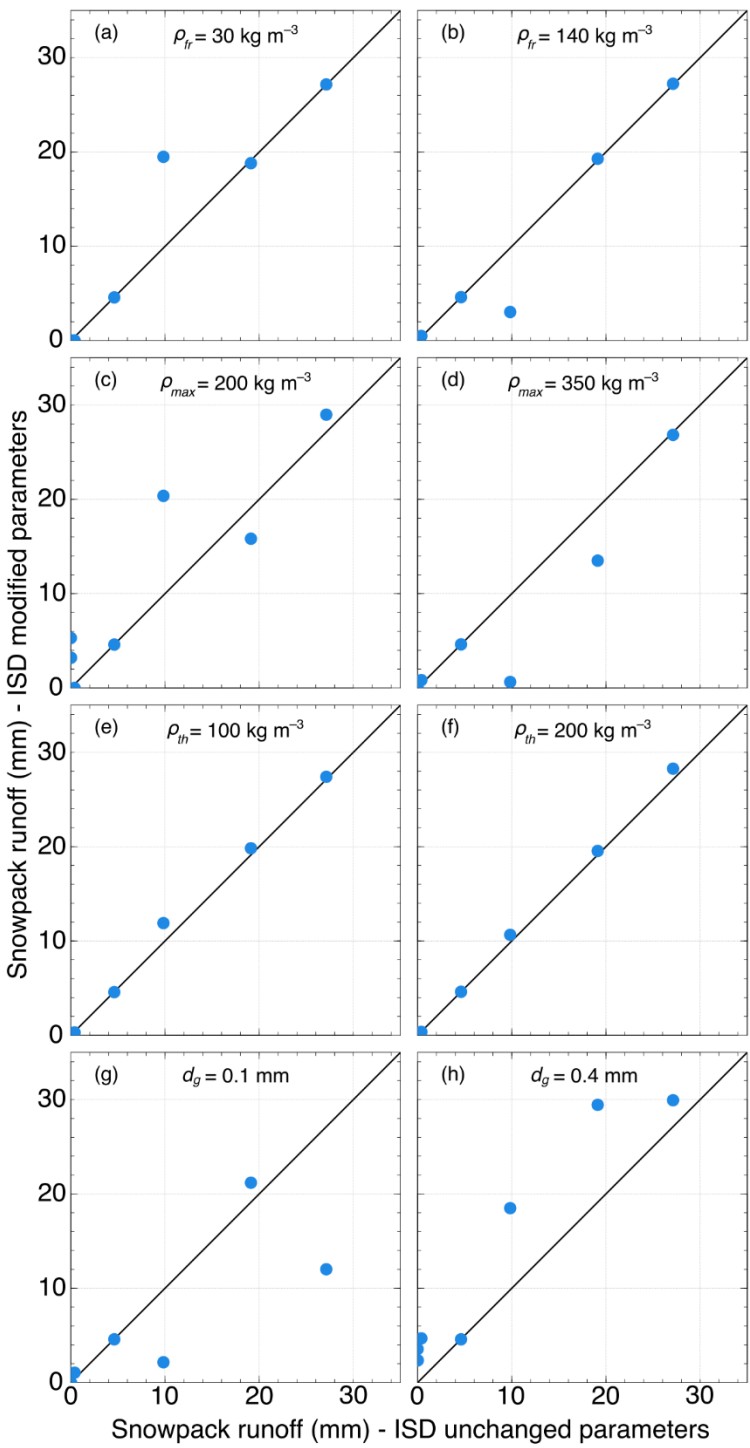

**Figure 14: Comparison of ROS–induced runoff simulated by SNOWPACK with ISD when the canopy snow parameters are unchanged (x–axis) and modified (y–axis). The modified parameter is identified at the top of each plot. The black line in all plots shows the 1:1 relationship.**



## 5 Discussion

### 5.1 Influence of ROS events on the sub–canopy snowpack

The wet and cold conditions that prevail in the boreal forest of eastern Canada cause the snowpack to form early in the season, thicken, and persist until late May or early June. Most ROS occurred in November–December, as temperatures from January
to March were generally too low for ROS events to take place. This contrasts with observations from the western United States and central Europe, where milder conditions are conducive to frequent ROS events from January to March for areas at elevations similar to our sites (McCabe et al., 2007; Hotovy et al., 2023).

Our ROS observations cover a range from small to very large events, and temperatures from below freezing to above 5°C (Fig. 4), providing a unique opportunity to systematically study the influence of ROS conditions on the sub–canopy snowpack. Air
temperature and amount of precipitation during a ROS event do not appear to be good predictors of melt–freeze layer thickness. This is because rainfall triggers the unloading of snow clumps and is further redistributed unevenly across the snow surface, leading to the formation of highly heterogeneous melt–freeze layers. In addition, the thickness of melt–freeze layers can decrease or even disappear over time due to gradient metamorphism (Domine et al., 2009), or increase as rainwater from subsequent ROS events accretes on the layer (Kapil et al., 2010). Overall, this complements the findings of previous studies,
i.e. that vegetation modulates the factors driving ROS runoff such as the snow water equivalent and the available energy for melting the snow (Storck et al., 2002; Wayand et al., 2015).

The cases presented in Fig. 6 suggest that preferential flow is an important transport mechanism for liquid water in the sub–canopy snowpack. These results are supported by the photos in Fig. 5, which demonstrate that ROS events may result in preferential flow under the canopy. The presence of percolation channels in the sub–canopy snowpack was also observed in
other studies (Bründl et al., 1999; Teich et al., 2019). Percolation channels are formed in cold snowpacks before matrix flow takes over as the snowpack warms and wet snow metamorphism prevails (Hirashima et al., 2019). Although percolation channels can form at any place in the snowpack due to boundary conditions (Schneebeli, 1995; Avanzi et al., 2016), it is more likely to find larger channels lower in the snowpack, as in Fig. 5c. This is attributed to larger grains such as faceted crystals and depth hoar that reduce the water–entry capillary pressure head of wetted snow layers (Waldner et al., 2004; Katsushima et
al., 2013). In general, water percolating through the snowpack by preferential flow accelerates the hydrological response to ROS events when compared to matrix flow (Singh et al., 1997; Waldner et al., 2004; Würzer et al., 2016). Since percolation channels were widely observed under the canopy, it suggests that snow conditions and properties under the canopy favor this water transport mechanism during ROS events and perhaps a rapid runoff response.

### 5.2 SNOWPACK simulations under a boreal canopy

Our results show that SNOWPACK generally overestimates the snow height below the canopy at both sites (Fig. 7). Except for winter 2018–19 at MF, the overestimation of snow height coincides with an overestimation of snow density (Figs. 7 and S8 to S15), leading to an even more overestimated snow water equivalent. At BRV, this could be attributed the accuracy of



ERA5-Land precipitation used as an input variable. In general, this could also be explained by too little interception simulated by SNOWPACK due to an underestimation of evaporation and sublimation by the model. This causes the storage capacity of the canopy to be reached too rapidly, thereby increasing snow accumulation on the ground. From 2018 to 2023 at MF, we simulated evaporation and sublimation losses ranging from 32 and 55 mm during the snow cover period. This corresponds to roughly 30% of what Isabelle et al. (2020) observed at the same site for the snow seasons of 2016–17 and 2017–18. We thus conclude that SNOWPACK underestimates mass loss of intercepted snow by as much as factor of 3, highlighting the need for further model development in this area.

SNOWPACK overestimates surface temperature during clear sky conditions. During daytime, the big–leaf radiative transfer might be at fault as direct incoming shortwave radiation breaching through the canopy is not simulated with this approach. During nighttime, an underestimation of snow surface layer density impacts thermal conductivity and the heat transfer between the air and the snow surface. Under cloudy conditions, SNOWPACK simulations of $T_{surf}$ improve because shortwave radiation is diffuse and because air and snow surface temperatures are similar. Our scores for snow density and grain types are inferior to those of simulations performed in alpine terrain (Lehning et al., 2001) or forest openings (Rasmus et al., 2007). This is partly explained by difficulties in sampling snow density and identifying grin types under trees due to the heterogeneous layering of the sub–canopy snowpack (Bouchard et al., 2022). Also, two–dimensional snow–forest processes that affect snowpack structure such as snow unloading and preferential canopy dripping, are difficult to capture in a one–dimensional model.

**5.3 Canopy parameterization, snowpack structure and runoff**

Using our intercepted snow densification scheme instead of the original module is a net gain in simulating the number and thickness of melt–freeze layers in the sub–canopy snowpack. Indeed, the unloading of denser snow consisting of small rounded grains creates fine–over–coarse transitions where liquid water is retained (Wever et al., 2016). In nature, melt–freeze layers are more likely promoted by a strong sub–canopy snowpack heterogeneity resulting from non–uniform processes such as unloading and meltwater dripping (Teich et al., 2019; Bouchard et al., 2022). However, our study suggests that simulating the canopy snow evolution helps reproduce the general features of melt–freeze formation under canopy (Table 6).

Delayed and reduced ROS runoff is an indirect consequence of simulating the unloading of denser snow of small rounded grains with ISD. However, in the absence of snowpack runoff measurements, we cannot directly validate this behavior and whether this constitutes an improvement in the simulated snowpack hydrological response to ROS events. The MF site is located within a catchment which the daily average discharge is continuously gauged at the outlet (station 051004 Des Aulnaies https://www.cehq.gouv.qc.ca/hydrometrie/historique_donnees/default.asp). Although the discharge station recorded an increase in streamflow from large ROS events of the study period, we refrain from drawing conclusions from these measurements for two reasons. First, the daily time step of discharge measurements is too coarse a resolution, as ROS–induced runoff was usually generated within one day of the rainfall event. Second, the forest cover in the catchment is discontinuous and snow properties are highly heterogeneous spatially so water transport mechanisms in the snowpack may differ significantly





under canopy and in forest gaps, as shown by Bouchard et al. (2022). Also, the relationship between snowpack runoff and discharge in streams should consider initial soil moisture, and the flow network (Wever et al., 2017).

Nevertheless, we compared our simulations with sprinkling water experiments in subalpine snowpacks by Singh et al. (1997) and Juras et al. (2017), who showed a runoff response to rainfall within an hour. This is faster than what we observe from temperature profile measurements (Fig. 6) or simulate with ISD (Fig. 12j) with a runoff response of several hours. Conditions

in those experiments such as a high rainfall rate, a shallow and warm snowpack, and the absence of canopy cover could explain the differences with our study.

### 5.4 Limitations

The first limitation of this work is the lack of measurements of intercepted snow properties in the canopy. Such measurements would have helped to accurately parameterize the age–based densification function. The sensitivity analysis shows relatively

low impact of canopy snow parameters on melt–freeze layers formation, but a larger influence on runoff that cannot be neglected (Fig. 14). In the past, observational studies have mostly focused on quantifying the mass of snow intercepted by the canopy (Friesen et al., 2015; Raleigh et al., 2022). Given the effect of canopy snow properties on ROS runoff, future studies should attempt to measured fundamental snow properties in the trees like snow density and *SSA*, and monitor canopy snow temperature to better understand snow metamorphism in the canopy.

A second important shortcoming is the lack of continuous snowpack runoff measurements, which limits our conclusions regarding the hydrological impact of ROS. Although soil–snow interface temperature provides information on the presence of liquid water at the base of the snowpack, it does not quantify the volume of water outflow. Lysimetric measurements should be used to that end despite technical and logistical challenges (Kattelmann, 2000; Floyd and Weiler, 2008; Webb et al., 2018b). Such measurements would allow to evaluate the performance of canopy snow densification on snowpack runoff.

This study is also limited by the single–point simulations of SNOWPACK, which contrast with the highly spatial heterogeneous character of forest snow (Bouchard et al., 2022). SNOWPACK, as any other multi–layer snow models to date, is not yet designed to reproduce the spatial heterogeneity of the snow cover from tree to tree. Therefore, further modeling developments are needed to better represent spatially variable vegetation–snow processes like interception and unloading (Vincent et al., 2018) or even radiation transfer (Jonas et al., 2020) in multi–layer snow models.

## 6 Conclusion

In this work, we first aimed to better document how rain-on-snow (ROS) events influence the sub–canopy snowpack structure through observations. To do so, we recorded nearly 50 ROS events, monitored the snow thermal regime and assessed snow properties and structure from snow pit measurements over multiple winters at two boreal sites representative of different bioclimatic areas of eastern Canada. We show that rain falling through the canopy before reaching the snow cover leads to the

formation of thick melt–freeze layers and that preferential flow is a major water transport mechanism in the sub–canopy snowpack.



Our second objective was to evaluate the multi–layer 1–D snow model SNOWPACK under a boreal canopy. Although designed to simulate alpine snow covers, SNOWPACK was found to be suitable for snowy and cold boreal environments such as those found in eastern Canada. It provides acceptable simulations of snow height and snow density, and accurately simulates the snow surface temperature. However, the observed melt–freeze layers resulting from ROS were generally not well simulated by the model.

The third objective of this study was therefore to assess the impact of implementing an age–based intercepted snow densification function in SNOWPACK on snowpack structure and runoff from ROS events. We obtained improved simulations of the number and thickness of melt–freeze layers. These improvements were obtained at both sites, illustrating the transferability of the function. The densification function also reduces and delays ROS–induced snowpack runoff as it produces dense and fine–grain layers of unloaded snow over lighter snow layers of large grains. The parameters of this function have a low impact on the simulation of melt–freeze layers. In contrast, these parameters were found to affect more strongly the simulated ROS–induced runoff, highlighting the need for documenting the physical properties of snow in the canopy.

In summary, our findings show that the evergreen canopy modulates snowpack structure, preferential flow, and snowpack runoff during ROS events. Our work is another step to better reproduce the properties of canopy snow and gives insights to further observational and modelling efforts in hydrology applied to snow–dominated forested environments. Studying the effect of canopy snow properties on runoff at larger scales would be a logical next step. Finally, the multi–year data set presented in this study can further be used for future model validation and improvement in a context of increasing winter rainfall events.

## Appendices

### Appendix A – SNOWPACK initialization file parameters

| Parameter | Value |
|---|---|
| Variables output time step (min) | 15 |
| Snow Profile time step (min) | 30 |
| Calculation time step (min) | 15 |
| Enforced measured snow height | False |
| SW forcing mode | Incoming |
| Atmospheric Stability | MO_MICHLMAYR |
| Roughness length (m) | 0.002 |
| Change Boundary conditions | False |
| Measured $T_{surf}$ | False |
| Change Boundary conditions | False |
| SNP SOIL | True |
| Soil flux | False |

Table A1: SNOWPACK.ini file parameters



**Appendix B – SNOWPACK parameterization of interception and unloading**

The temporal change of canopy storage ($dI/dt$; mm day$^{-1}$) is a function of interception rate ($\Delta I$; mm day$^{-1}$), evaporation and
sublimation of intercepted snow ($E_{int}$ mm day$^{-1}$), and liquid or solid unloading from the canopy ($U$; mm day$^{-1}$):

$$\frac{dI}{dt} = \Delta I - E_{int} - U \ . \tag{A1}$$

The interception rate is calculated following Hedstrom and Pomeroy (1998):

$$\Delta I = c_u (I_{max} - I)\left(1 - e^{-\left(\frac{(1-c_f)P}{I_{max}}\right)}\right) , \tag{A2}$$

where $c_u$ (–) is an unloading coefficient, set to 0.7 as suggested by Pomeroy et al. (1998), $I$ is the mass initially stored in the
canopy (mm), $c_f$ (–) is the gap fraction which takes a value between 0 and 1, and $P$ (mm day$^{-1}$) is the precipitation. $I_{max}$ (mm)
is the maximum capacity of the canopy (Schmidt and Gluns, 1991). When canopy interception is in the liquid phase , $I_{max}$ takes
the constant value of 0.3 mm m$^{-2}$ as suggested by Gouttevin et al. (2015). Otherwise, $I_{max}$ is estimated from the $LAI$ (m$^2$ m$^{-2}$),
the density of intercepted snow ($\rho_{s,int}$; kg m$^{-3}$) and a tree species–dependant factor, $i_{max}$ (mm):

$$I_{max} = i_{max}\left(0.27 + \frac{46}{\rho_{s,int}}\right) LAI \ , \tag{A3}$$

where $\rho_{s,int}$ is a polynomial function of $T_a$ in °C, $RH$ (0 – 1), $WS$ in m s$^{-1}$ and $T_{surf}$ in °C:

$$\rho_{s,int} = \alpha + \beta T_a + \gamma T_{surf} + \delta RH + \eta WS + \varphi T_a T_{surf} + \mu T_a WS + \nu RHWS + o T_a T_{surf} RH \ , \tag{A4}$$

where $\alpha = 90$, $\beta = 6.5$, $\gamma = 7.5$, $\delta = 0.26$, $\eta = 13$, $\varphi = -4.5$, $\mu = -0.65$, $\nu = -0.17$ and $o = 0.06$. $\rho_{s,int}$ is limited to values
between 30 and 250 kg m$^{-3}$.

Unloading is the difference between the canopy storage and $I_{max}$ over each time step and happens when the storage exceeds
the maximum capacity of the canopy:

$$U = \frac{\max[0, \ I - I_{max}]}{\Delta t} \ . \tag{A5}$$

The throughfall is calculated as follow:

$$T = P - \Delta I + U \ . \tag{A6}$$

In this parameterization, unloading and precipitation are merged before being added as a new snow layer to the snowpack.
Solid or liquid unload from the canopy takes therefore the same properties as solid or liquid precipitation.



*Code and data availability.*    Documented code of SNOWPACK version 3.6.0 is available on GitLab (https://gitlabext.wsl.ch/snow-models/snowpack). Data from snow pit observations, the monitoring stations and observed rain-on-snow events are freely available at 10.5281/zenodo.10357450.

*Author contributions.*   BB designed the study with DN, FD and ML. BB collected and treated field data. PEI gap–filled and
provided in situ forcing data from Montmorency Forest. BB performed simulations with the model. NW, AM and BB made the modifications to the model. BB analyzed the results and wrote the manuscript with insights and feedback from all authors.

*Competing interests.*   Florent Domine is a member of the editorial board of The Cryosphere.

*Acknowledgements.*   The authors would like to thank the staff of the Montmorency Forest for their logistical support during the field visits. We also thank André Desrochers for lending us his snowmobiles from 2018 to 2020, and Charles Villeneuve
for ensuring their maintenance and preparing the snowmobile trails before our visits. The authors also thank Éric Boucher and Christian Juneau and Martin Lapointe for their help in preparing, setting up and maintaining the monitoring stations. We thank Antoine Thiboult for providing the forcing data set at the Bernard River Valley site. We also thank all the people who accompanied Benjamin Bouchard in the field to dig snow pits, especially the members of PÉGEAUX and the graduate students, post-docs, and research associates from the Civil and Water Engineering Department of Laval University. We thank the staff
and scientists of the Laboratory of Cryospheric Sciences (CRYOS) at EPFL (Lausanne) and the SLF (Davos) for hosting Benjamin Bouchard for summer research stays in 2018 and 2022. We thank the Sentinel North student mobility grant program, the CentrEau mobility program, and the International Office of Laval University for supporting the research stay at the SLF in 2022. Benjamin Bouchard's work was supported by the Natural Sciences and Engineering Research Council (NSERC) and the Sentinel North program. The authors acknowledge the financial contribution of Environment and Climate Change Canada
through the Grants & Contributions program (projects #GCXE20M016 and #GCXE22M013) and the Cold-region climate projection for hydrological applications (EVAP-2; project #ALLRP 549108 - 19).

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
