# Peer review of "Impact of intercepted and sub—canopy snow microstructure on snowpack response to rain-on-snow events under a boreal canopy"

_EGUsphere, 2023_

## Author Comment (AC1)

**EGUSPHERE-2023-3012 - Impact of rain-on-snow events on snowpack structure and runoff under a boreal canopy**

**Responses to Dr. Giulia Mazzotti**
* * *
First of all, we wish to thank Dr. Giulia Mazzotti for providing constructive and insightful comments and suggestions. Based on these, we propose changes to the original manuscript. We hope that these changes, if accepted, will make the story clearer and the paper easier to read. In particular, we have carefully revised the Methods (observational data and SNOWPACK modeling) and the Discussion sections, but we also suggest changes to the Introduction and Results sections. Our answers below are in blue, whereas excerpts from the manuscript are in blue *italics* with modifications in **bold**. The DOI of the newly cited references in the response letter are presented at the end of the document.
* * *
**General comments:**

The study 'Impact of rain-on-snow events on snowpack structure and runoff under a boreal canopy' by Bouchard et al. explores how rain-on-snow (ROS) events impact sub-canopy snowpack structure based on experimental findings, and further assesses whether the physics-based, multi-layer model SNOWPACK succeeds in capturing these effects. Improvements to the representation of canopy snow in SNOWPACK are also suggested. I really enjoyed reading this manuscript. It addresses multiple topics that are known to be important and understudied in the forest snow (modelling) community, and I commend the authors for the laborious and extensive field campaigns – such datasets are unique and highly valuable for the community. The manuscript is well written and generally well structured. The methods are very clear and easy to read (although some additional details are needed, see below). The results include many interesting findings but would benefit from some streamlining (potentially just an introductory paragraph). In terms of content and logic, I identified some issues that should be addressed before the study can be considered for publication, see specific comments below. The suggested changes mainly aim at making the argumentation easier to follow, do not question the novelty and value of the study, and I don't see any of them being problematic to tackle. I am confident that this article will make for a great contribution to the literature following revisions. I encourage the authors to reach out if they have questions to my comments and wish to discuss any of them further (giulia.mazzotti@meteo.fr).

**Specific comments:**

**1. Study goals (L 94 ff):** I feel that the current way of introducing the study objectives does not do justice to the interesting research presented thereafter. The statement 'this study aims to help fill the research gap on ROS events in the boreal forest' is very vague. Then, it is suggested that modelling is used as a tool to help field data interpretation. Then again, the third objective addresses a model improvement with a non-obvious link to ROS (and remains somewhat disconnected from the two former objectives). And finally, 'improved understanding' is stated as overall goal, which disregards the model development work.

You could make a much stronger case by better connecting the three aspects of the study and by highlighting the model testing/development as part of the overall objective. I suggest arguing somewhat along these lines: 'This study addresses observation and modelling of ROS events in the boreal forest. Specifically, we first use field observations […]. These field observations provide the unique opportunity to evaluate the multi-layer, microstructure resolving model SNOWPACK […]. Motivated by model shortcomings identified in this evaluation (or: motivated by the hypothesis that unrealistic properties of unloaded snow hamper the performance of SNOWPACK in simulating how sub-canopy snowpacks react

to ROS events), we finally suggest an alternative representation of canopy snow properties and assess its impact on SNOWPACK simulations […]. Altogether, this work improves both our understanding of ROS events in forests and our ability to capture their impacts on snowpack structure and runoff by physically-based models (which is important as such events are expected to become more frequent).' That's just a suggestion, I hope you get the idea…

Thank you for this comment. We agree that a better link between the objectives would provide a clearer narrative for the study. Therefore, we suggest the following changes to the manuscript:

l. 89 to 98 (Introduction): "*In this study, **we address ROS events in the boreal forest through observations and modeling. Specifically, we first use** field observations to document the ROS–induced alterations of sub–canopy snowpack structure, with a focus on melt–freeze snow layers and preferential flow channels. This objective builds on the work of Bouchard et al. (2022), who found that the sub–canopy snowpack is highly heterogeneous and has high permeability, **expected to** facilitate downward water flow. **These field observations provide the unique opportunity to evaluate the one-dimensional, multilayer, microstructure resolving model SNOWPACK in the boreal forest of eastern Canada. Motivated by the hypothesis that unrealistic properties of unloaded snow hamper the performance of SNOWPACK in simulating how sub-canopy snowpacks react to ROS events, we suggest an alternative representation of canopy snow properties and assess its impact on SNOWPACK simulations of melt-freeze layers** and ROS–induced runoff. Overall, **this work improves both our understanding of ROS events in the boreal forest and our ability to capture their impacts on snowpack structure and runoff thanks to physically-based modeling. This is especially important as ROS events are expected to become more frequent in the future.*"*

**2. Logic of linking ROS and canopy snow:** While the title suggests a focus on rain on snow events, the modelling part focuses on the representation of canopy snow. The authors seem to implicitly assume that the two processes are interlinked., i.e., that canopy snow and its unloading are what makes the impact of ROS in forests different to their impact outside of forests. While this might be correct to some extent, it is not intuitive/self-explanatory, and not yet sufficiently justified in the study. This aspect needs to be worked out better, and I think there are multiple components to it.

> a) Experimental evidence: it needs to be stated clearly that the snowpack features of the sub-canopy snowpack that favor runoff from ROS events are not present in canopy gaps (or open areas) and result from interception and unloading. This is implicitly alluded to in the paragraph starting L 54 but including a short summary of the main differences between sub-canopy snowpacks and the snowpack in canopy gaps found by Bouchard et al. (2022) would make this link more explicit.

A reminder of the differences in gap and subcanopy snowpack structure observed by Bouchard et al. (2022) would indeed be relevant here. We suggest adding a few sentences in the Introduction to make it clearer that the subcanopy snowpack has features that promote downward water flow that are not observed in forest gaps. We also suggest removing the reference to Colbeck (1983) and replacing it with references to Musselman et al. (2008) and Molotch et al. (2016), which are more relevant in this context. The proposed changes are as follows:

l. 56 to 61 (Introduction): "*A thin snowpack below the **canopy increases the vertical temperature gradient, favoring** gradient metamorphism and grain growth **(Musselman et al., 2008; Molotch et al., 2016)**. **Confirming this, Bouchard et al. (2022) observed that the snow density under the canopy is lower and the grains are larger and have a lower SSA than for the snowpack within the adjacent forest gaps. The combination of a lower density and a lower SSA leads to a greater permeability of the subcanopy snowpack, facilitate the downward water flow compared to the gaps.** Previous studies have*

*also shown that preferential flow is more likely under the canopy due to canopy* snow unloading, meltwater dripping, and accumulation of vegetation debris **over the sub–canopy snow cover** *(Bründl et al., 1999; Teich et al., 2019).*"

    b)   Causality: The title suggests that the study addresses how ROS events affect the snowpack structure, but a big part of the study is about how a snowpack structure shaped by interception and unloading affects the snowpack's reaction to ROS. By not acknowledging this distinction, the two processes become somewhat entangled. It might be worthwhile to reconsider the title (although I don't have a concrete suggestion).

This is a good point. We suggest the following title:

l. 1 to 2 (Title): "***Impact of intercepted and sub-canopy snow microstructure on snowpack response to rain-on-snow events*** *under a boreal canopy.*"

    c)   Model choices and structure in SNOWPACK: one of the reasons why the above-mentioned causality is confusing is that in SNOWPACK, rain on snow and unloading events may indeed be entangled, but this may be a consequence of some model choices that are quite unusual; in particular, the fact that unloading occurs only when interception capacity is exceeded (as far as I know, most common unloading parametrizations include some sort of time decay function to remove canopy snow even if interception is below capacity). In Gouttevin et al., it is stated that this choice allows for a gradual unloading. However, a maybe less evident consequence of this is that ROS events induce big unloading events because the onset of rain entails a large drop in interception capacity (I hope I am interpreting this correctly, but it seems to follow from what you write in L 657 'when canopy interception is in the liquid phase', and it agrees with Fig 10). This could be why ROS and unloading become linked. I wonder how much your results would change if unloading was triggered differently in SNOWPACK (e.g. by wind, when capacity is not yet reached, or using other approaches as summarized e.g. by Lumbrazo et al. 2022). I am not saying that you need to test alternative parametrizations, but I think that this particularity of SNOWPACK should be addressed in the discussion to facilitate interpretation of your results, and you should acknowledge that your results are somewhat specific to SNOWPACK (probably in Section 5.3). In the same section, I suggest you also comment on the fact that the unloading implementation in SNOWPACK still lacks some processes that would likely impact the subcanopy snowpack (e.g. drip unloading?).

This is a good remark and your interpretation is correct. A large drop in the interception capacity does indeed occur when the precipitation is liquid. Therefore, the model often simulates large snow releases during ROS events. This unloading parameterization is in fact specific to SNOWPACK. Other parameterizations, e.g. drip-based (e.g. Andreadis et al. (2009)) or wind-based (Roesch et al. (2001)), would undoubtedly give different results in terms of snow cover structure. Given the conditions prevailing at our site, we would not expect drip unloading to trigger frequent unloading events at our sites, as warm conditions without precipitation are rarely observed from October to March. Wind unloading, may trigger more frequent unloading during winter, which would reduce the snow storage in the canopy. We suggest the following addition to the manuscript:

l. 601 (Discussion): "***Our modeling results are specific to the unloading parameterization used in SNOWPACK (see eq. A5), which promotes large snow unloading during ROS events. Lumbrazo et al. (2022) showed that the choice of unloading parameterization affects the timing of snow unloading, as well as the canopy mass removal from unloading and sublimation. It is likely that the unloading parameterization would also influence the snow microstructure and ROS-induced runoff simulations.***"

d) Comparison to non-forested sites: Since Bouchard et al. 2022 and 2023 have observations in forest gaps, it would have been really cool to see what the model does there (i.e. to collect further evidence that it is indeed the canopy snow that makes the difference). The study in its present form is already rich enough and I am aware that conducting such simulations would be a lot of additional work, so I am not asking to add this to the revision, but if you plan to expand on this work in the future, I think this would be the perfect next step (consider mentioning in outlook?). I know that SNOWPACK is not yet suited to simulate snow in canopy gaps, but I think that the approach used here: https://doi.org/10.5194/egusphere-2023-2781 could easily be applied to SNOWPACK as well.

We agree with Dr. Mazzotti that a comparison with forest gaps would allow a deeper interpretation of the role of the forest canopy on snowpack structure and runoff during ROS events. Based on observations presented in this manuscript (see Fig. 11a), Bouchard et al. (2022) found that ROS events lead to the formation of thin and clear ice layers in small forest gaps, instead of thick melt-freeze layers under the canopy. Therefore, the canopy cover affects the snowpack structure during ROS events. Simulating the evolution of the snow microstructure in forest gaps with SNOWPACK would allow to further interpret results of Bouchard et al. (2022 and 2023), especially with respect to the water transport mechanisms in the snow during ROS events (matrix versus preferential flow). The application of the approach of Mazzotti et al. (2023) [doi: 10.5194/egusphere-2023-2781] would be particularly relevant here. However, this goes beyond the objectives of the study. We suggest modifying the final paragraph of the conclusion to mention this as an outlook.

l. 639 to 644 (Conclusion): "*Our work is another step **towards** better reproduc**ing canopy snow properties of and **provides** insights **for** further observational and modelling efforts in hydrology applied to snow–dominated forested environments. **Investigating** the effect of canopy snow properties on runoff at larger scales, **developing a robust methodology to assess canopy snow metamorphism from observations, and coupling detailed canopy structure schemes to multi-layer snow models to simulate snow microstructure in forest gaps** would be logical next steps. **The approach recently developed by Mazzotti et al. (2023) looks promising for this purpose.** Finally, the multi–year data set presented in this study can further be used for future model validation and improvement in a context of increasing winter rainfall events.*"

**3. ISD implementation – motivation, interpretation of results, discussion:** The implementation of canopy snow densificaty (and microstructure) evolution is doubtlessly a nice contribution addressing a need that has been highlighted multiple times in the literature and could be valorized more.

a) At the beginning of sectionn 3.2.1, instead of stating 'as part of our third research objective', I would recall the reasoning behind this development (i.e, currently unrealistic representation of unloading snow properties). In particular, I would note that we expect a microstructure resolving, multi-layer model like SNOWPACK to be more sensitive to the simplified representation of canopy snow than 'simpler' snow models these parametrizations were originally intended for.

We agree that the reasoning behind our model development strategy and the use of SNOWPACK should be strengthened. We suggest the following modification:

l. 223 to 224 (Methods): "***We expect SNOWPACK simulations under the canopy to be sensitive to the properties of the unloaded snow. In its current form, SNOWPACK does not simulate the evolution of snow properties in the canopy. Therefore,*** we implemented **a canopy snow parameterization that accounts** for the **of the density and microstructure of the intercepted snow (Fig. 2)**."

The results section left me with some open questions that should be addressed (or maybe you just need to guide the reader a bit more…):

b) Figures 10 and 12: I am a bit surprised that the density of the unloaded snow with the IM version is always so low. After all, density does depend on temperature, and the Appendix states that it can reach up to 250kg/m3. Since most unloading (and especially the example shown in Figure 12) occurs during ROS, I assume this to coincide with a rather high air temperature so I would have expected a higher density, too. Do you have an explanation for this? Maybe it would help to describe more in detail what happens in the model when snow unloads at the onset of a ROS event.

Equation A4 is parameterized so that there are always terms on the right-hand side of the equation that offset others:

$$\rho_{s,int} = \alpha + \beta T_a + \gamma T_{surf} + \delta RH + \eta WS + \varphi T_a T_{surf} + \mu T_a WS + \nu RHWS + o T_a T_{surf} RH$$

where α = 90, β = 6.5, γ = 7.5, δ = 0.26, η = 13, φ = –4.5, μ = –0.65, ν = –0.17 and o = 0.06. This results in a fresh snow density that almost never reaches the upper limit of 250 kg m$^{-3}$. For example, in the case of the snow unloading event shown in Fig. 12, the variables of eq. A4 take the following values:

$T_a$: –3.43 °C
$T_{surf}$: –3.82 °C
$RH$: 95.99 %
$WS$: 1.53 m s$^{-1}$

This results in a density of fresh snow of 78.86 kg m$^{-3}$. In that case, the 1$^{st}$, 4$^{th}$, 5$^{th}$, 7$^{th}$, and 9$^{th}$ terms are positive, while the others are negative. We suggest the following modification to the manuscript to provide a clearer explanation of snow unloading during ROS events with the initial version of the canopy module:

l. 218 to 220 (Methods): "*Unloading occurs when canopy storage exceeds the maximum storage capacity of the canopy.* **At the onset of a ROS event, snow in the canopy unloads first and** *contributes to the formation of a new snow layer on top of the snowpack* **with fresh snow properties based on the weather conditions at the time of unloading.**"

c) In Section 4.3.2: Is the formation of MF layers that are detected by ISD but not by IM always preceded by an unloading event? I would specify this.

All modeled melt-freeze layers with ISD resulting from ROS events from October through March were preceded by snow unloading. This is not surprising given that the presence of liquid precipitation explains virtually all cases of large drop in interception capacity observed in winter. In fact, as mentioned earlier, cases of reduced interception capacity associated with warm but dry conditions are very rare. We suggest the following addition to the manuscript:

l. 452 (Results): "**All ROS-induced melt-freeze layers correctly reproduced with ISD from October through March were preceded by snow unloading.**"

Related: did you look at the impact of unloading snow at the snow surface also when there is no ROS?

SNOWPACK simulated small snow unloading events without liquid precipitation when interception storage slightly exceeded the maximum capacity of the canopy. With ISD, this generally resulted in the formation of a thin layer of small rounded grains that eventually became buried and experienced snow metamorphism into faceted crystals within the snowpack. We did not note instances where these buried layers impeded water percolation from ROS later in the season. This could be explained by snow metamorphism reducing the fine-over-coarse layer transitions that are generated by dense snow unloading. We suggest mentioning this in the revised version of the manuscript.

l. 452 (Results): "*However, not all simulated unloading events were followed by a ROS events. We did not note instances where these buried layers impeded water percolation from ROS later in the season as snow metamorphism reduces the fine-over-coarse layer transitions.*"

    d) In Section 4.3.3: You suggest that more preferential flow and snowpack runoff occurs with IM than with ISD. Wouldn't this match your observation of preferential flow channels better? The decreased runoff with ISD seems contradictory to Bouchard 2022 who states that the subcanopy snowpack structure should favor rapid runoff?

This is correct. In general, the ISD model reduces and delays preferential flow in the snowpack compared to the IM model. This is consistent with our observations of the temperature profile evolution shown in Fig. 6, where water appears to reach the lower snow layers by preferential flow within 1 or 2 days instead of a few hours as simulated by IM. Without lysimetric measurements, we cannot confirm that our simulations with ISD better reproduce snowpack runoff from ROS events than those with IM. Bouchard et al. (2022) stated that the subcanopy snowpack would promote a faster runoff response than the gap snowpack. This hypothesis could only be strengthened by complementary simulations in forest gaps or confirmed by field observations of water flow through the snowpack under canopy and in adjacent forest gaps.

    e) Section 4.4: The large sensitivity to dg left me wondering whether the real impact of your model adaptation really comes from the densification, or whether it is from the treatment of snow microstructure descriptors. My understanding is that while densification is a continuous function (actually, also in IM), the treatment of canopy snow grain type/size is binary (in ISD, while in IM they always get the same properties, see Fig 2?). I am not familiar with the relationship between water content, saturation threshold for preferential vs. matrix flow, and snow properties (density, grain size?). From a quick look at Wever et al. I get that grain radius is the key parameter. Now my question: does the modification of density relative to IM really make the difference, or was the real problem in IM that the unloading snow was attributed fresh snow properties? I think it's crucial to explore and clarify this (either way it's an interesting finding – you identified an important shortcoming of the original implementation!).

This is a good observation. Indeed, based on eq. 2 from Wever et al. (2016), the grain size controls the water entry pressure, which affects the transition of water from the preferential to the matrix flow domain. This explains why snowpack runoff is highly sensitive to $d_g$. However, as shown in Fig. 14, snow density (especially the maximum density of intercepted snow, $\rho_{max}$) also influences the ROS-induced runoff. This is due to the Van Genuchten parameters, which combine $\rho_s$ and $d_g$ (see eq. 6 from Wever et al. (2015)) so that the capillarity increases with smalelr grain size and higher density. This is also because the hydraulic conductivity ($K_s$) in SNOWPACK is parameterized from the relationship of Calonne et al. (2012) (see eq. 14 of Wever et al., (2014)) who estimate $K_s$ from $\rho_s$ and $d_g$. Overall, we agree with Dr. Mazzotti that this should be clearly mentioned in the Discussion. We suggest the following changes in the manuscript:

l. 586 to 588 (Discussion): "*Delayed and reduced ROS runoff is an indirect consequence of simulating the unloading of denser snow of small rounded grains with ISD.* **The greater effect of grain size on runoff sensitivity than snow density (Fig. 14) can be explained by the parameterization of the water entry pressure of the snow layers, which drives the transition between the preferential and matrix flow domains (Wever et al., 2016). Since the hydraulic conductivity and the Van Genuchten parameters are estimated from $\rho_s$ and $d_g$ in SNOWPACK (Wever et al., 2014; Wever et al., 2015), snowpack runoff is also affected by the density of unloaded snow. This sheds light on the hydrological influence of microstructural descriptors of intercepted and unloaded snow.** *In the absence of snowpack runoff measurements, we cannot directly validate* **the ISD parameterization** *and whether this constitutes an improvement in the simulated snowpack hydrological response to ROS events*."

**Technical comment:**

Most of these questions/suggestions are aimed at improving readability / eliminate unclear phrasing and typos.

**4.** L 20: lead to the formation (without s)

Thank you. This will be corrected.

**5.** L 28: Use of 'indeed' is awkward. 'Specifically'? 'In fact'?

Thank you. We will replace "indeed' by "in fact".

**6.** L 30: 'Our results show …' – this statement will need revision. To really show this, you would need to present data of a snowpack outside of a forest as well.

This is a good point. We will replace "our results show" by "our results suggest".

**7.** L 66: 2015 is no longer 'recently' (unfortunately! ;-) )

Indeed. We will remove "recently".

**8.** L 76: 'This is partly due to limited field data' – a bit too vague. What data? And do you mean 'data to inform model improvements'?

Thank you for your comment. We suggest clarifying the statement as follows:

l. 76 to 77 (Introduction): "*This is partly due* **to a lack of calibration and validation snow data in forests,** *and* **to** *interception parameterizations derived from a few observational studies that are difficult to generalize to other climates (Lundquist et al., 2021).*"

**9.** L 84: 'can undergo metamorphism before being unloaded' – this would be a good place to note that these interception and unloading parametrizations were originally intended for models that are much less complex than SNOWPACK and do not resolve snow microstructure, see my comment above.

Thank you. In accordance with our response from comment 8, we suggest the following changes to the manuscript:

l. 83 to 86 (Introduction): "*However, snow can remain in the canopy for up to several weeks (MacDonald, 2010; Lumbrazo et al., 2022) and undergo metamorphism before being unloaded. **Since these parameterizations do not account for snow metamorphism,** this can lead to inaccurate simulations of the physical properties and stratigraphy of the sub–canopy snowpack after an unloading event. **This,** in turn**,** could alter the simulated downward water flow from a ROS event.*"

**10.** L 119: 'when estimated' – maybe worth mentioning the method (just say if that was laser altimetry, visual inspection, photogrammetry?'). Same for BRV site.

We used a clinometer at both sites. This will be corrected.

**11.** Table 1: I would move this to the Appendix, it's very technical. But that's a matter of taste.

We agree to move this table to the Appendix.

**12.** L 138: 200km is a lot!

Indeed, but these are Canadian-scale distances. Sept-Îles is the closest "large" town to the site, which has an airport and a federal weather station.

**13.** L 143: The big difference in LAI is surprising. The hemispherical images look quite similar, and the trees are taller in the case of BRV. Could it be due to the method used to derive these values? What does this imply for the use of detailed snowpack models?

This is a good observation. The sky view fraction of hemispherical photos from the MF site in Fig. 1 is 0.10, which is lower than that of the BRV site (0.16). Given the variability of LAI values from measurements at both sites ($\pm$ 1.6 m$^2$ m$^{-2}$ at MF and $\pm$ 0.7 m$^2$ m$^{-2}$ at BRV) and the uncertainty of both instruments, which is on the order of 15 to 20 % (Fang et al. 2019), it is possible to obtain such a difference.

In SNOWPACK, the LAI is used to estimate the maximum interception capacity of the canopy, the absorption of shortwave radiation by the canopy, the contribution of the canopy to longwave radiation, the aerodynamic resistance, and the biomass heat flux. All of these factors are reduced when the model is run with small LAI values.

**14.** L 186: It is unclear to me whether this was done based on the data measured at the sites (i.e., the automated snow under the canopy measurements), the time-lapse cameras, the data at the stations, or a combination of all. Please be more specific.

This is a good point. We suggest the following modification to the paragraph:

l. 183 to 186 (Methods): "*In this study, we define a ROS event as at least 3 mm of rain **cumulated** over 12 h or longer while a minimum of 3 cm of snow on the ground is observed. It is one of the many definitions of a ROS event in the literature (Brandt et al., 2022)**, which** we chose for its simplicity**,** in the absence of runoff measurements. **Rainfall amount and duration were taken from the DFAR at MF, and from ERA5-Land reanalysis at BRV (see Sect. 3.4), whereas precipitation phase and snow height were respectively obtained from time-lapse images and ultrasonic measurements at both sites.** Note that we focus on ROS events that occur between November and March exclusively.*"

**15.** L 187ff: Here, I think you would do your storyline a favor by justifying why you chose to use SNOWPACK (and highlighting that your data offers an excellent opportunity to evaluate and further improve it for forest applications).

This is a very good suggestion, thank you. Here are the modifications that we suggest:

l. 189 to 193 (Methods): "***The SNOWPACK model was selected for several reasons. First,*** *it accounts for the processes that drive snow metamorphism and provides a complete* ***and detailed*** *representation of snow microstructure, thermal profile, snow settlement, and mass and energy balance* ***(Bartelt and Lehning, 2002; Lehning et al., 2002a; Lehning et al., 2002b). Second, it solves Richards' equation to represent matrix and preferential flow in the snowpack (Wever et al., 2016). Third, SNOWPACK includes a two-layer canopy module that simulates the exchanges of mass and energy between the vegetation and the underlying snowpack (Gouttevin et al., 2015). Our detailed snow dataset provides an excellent opportunity to validate and further improve this model for forest applications.***"

**16.** L 202ff.: It would be helpful to add a few words on what snow properties are used to determine this saturation threshold and therefore the transition from preferential to matrix flow, see my major comment #3. I think this will facilitate interpretation of Fig. 12 and make the justification of why you use a higher threshold than Wever et al. more convincing.

We agree with Dr. Mazzotti that more details about the saturation threshold parameter would help the reader interpret our results and model the choice of a higher threshold. We suggest the following changes to Section 3.1:

l. 201 to 203 (Methods)**:** "*In the model, this is conceptually characterized* ***by a threshold of liquid water content that marks the saturation*** *of the preferential flow domain ($\theta_{TH}$; $0-1$).* ***In practice, the use of $\theta_{TH}$*** *controls the movement of water from preferential to matrix flow. This allows for* ***water spreading at microstructure transitions, which leads to the formation of ice or melt-freeze layers, as*** *phase change is only possible in the matrix flow domain****.***"

l. 206 to 210 (Methods)**:** "***We tuned $\theta_{TH}$ based on SNOWPACK simulations at an open site, i.e. without vegetation effects, where we examined snowpack wetting after ROS. The simulations were compared with snow pit observations at the open site, where thin melt-freeze and ice layers resulting from ROS events were identified (unpublished data). These were better reproduced by simulations with a $\theta_{TH}$ greater than 0.35. Therefore, we set $\theta_{TH}$ to 0.35 in our modeling setup.***"

**17.** L 209: 'not shown': consider putting in supplementary material if you feel this is useful for readers (not a requirement)

Since we have provided some clarification about $\theta_{TH}$, we do not feel that showing the results of this analysis is useful for readers. In addition, since the simulations are compared to unpublished data from an open site, showing these results would make the manuscript much heavier.

**18.** L 210: 'geometric mean': it is unclear of what, and I am unclear about the purpose of providing this information. Of the hydraulic conductivity of the two layers? To get hydraulic conductivity at the nodes? Why would you need that? Please rephrase to make this clearer.

Indeed, providing this information directly in the text is not useful for the understanding of the study. We suggest deleting the sentence on l. 210-211 and move this to the table in Appendix A.

**19.** L228: This sounds as if Koch et al. derived this formula for this exact purpose, but that's not the case. I think you should give some context to avoid misunderstandings (something like: 'we use the formula from Koch et al., originally developed for … based on data from …')

This equation was originally developed to estimate snow thickness together with SWE derived from a GPS signal. We propose the following addition to the manuscript:

l.227 to 228 (Methods)**:** "*...based on Eq. 16 from Koch et al. (2019),* **which was originally developed to estimate snow height along with snow water equivalent measurements from an alpine site in Switzerland***:*"

**20.** Table 2: Heading 'For' seems a typo, please revisit. It would further be useful to add symbols to the variable names (e.g., I think that the 'direct throughfall fraction' is the gap fraction cf (–) in Eq. A2, but it would be good to make this unequivocal).

Thank you. The typo will be corrected. The symbols in Gouttevin et al. (2015) will be added to the table and defined in the text. The direct throughfall fraction and the gap fraction refer to the same parameter. We will also replace the gap fraction in Appendix B by the direct throughfall fraction.

**21.** L294: Does this mean that the phase determined from the images replaces the phase determined with the dual threshold? If yes, why do you need the dual threshold? Please clarify.

Since the phase of the precipitation affects the maximum interception capacity, it must be defined at each time step even, when if no precipitation is occurring. That is why we used a dual threshold in the first place. Then, when precipitation was observed, we validated the phase with the timelapse images and replaced the dual threshold value with either 0 (snow) or 1 (rain). We suggest the following changes to the manuscript to make this clearer:

l. 293 to 295 (Methods)**:** "**Since the phase of the precipitation is used to estimate the canopy interception capacity (see Appendix B)***, a linear transition with dual temperature thresholds at 0 and 2°C was first* **applied** *to define the phase of precipitation at any time step. Then, th**is** phase was* **validated using** *time–lapse images.*"

**22.** Figure 2:  For IM, is rho_new the same as rho_s,int in L 661 / A4? I guess so, but I would advise you to be consistent with the symbol because it can get confusing really quickly…

Gouttevin et al. (2015) use $\rho_{s,int}$ in the canopy interception capacity equation (Eq. A3), but in fact this variable is actually calculated as the density of new snow. We agree that this becomes confusing in our paper since we introduce the density of intercepted snow in the ISD model. We suggest replacing $\rho_{s,int}$ with $\rho_{new}$ in Eqs. A3 and A4 to be consistent with Figure 2.

**23.** Table 3 is not referenced anywhere in the text. I would mention here that you do a sensitivity analysis, so that Section 4.4 comes as less of a surprise.

We propose to add a paragraph to explain the objectives of our sensitivity analysis in the Methods section:

l. 306 (Methods): "**Given that** $\rho_{fr}$**,** $\rho_{max}$**,** $\rho_{th}$ **and** $d_g$ **values can be subject to site dependencies, we evaluated the sensitivity of the ISD module to these canopy snow parameters. Table 3 shows the low**

*and high values used in the sensitivity analysis, as well as the values assigned for the baseline analysis. For the sensitivity analysis, we manually changed the values of one parameter at a time to isolate its effect on the simulation results. We performed the sensitivity analysis exclusively for the winter 2018-19 at MF, as this is the year with the most snow pit observations. We first objectively assessed the melt-freeze layers formation, as described in Sect. 3.2 from Quéno et al. (2020). Briefly, this method evaluates whether the observed layer is well simulated ("hit") and whether the layer thickness is adequately reproduced. The simulated and the observed layer height must agree within a margin of 20% of the observed total snow height to be considered as a "hit". The observed melt-freeze layer thickness is well reproduced when the simulated layer thickness is half to twice that of the observed layer. We also evaluated the sensitivity of SNOWPACK to ROS-induced runoff, although no field data were available to validate this process.*"

**24.** Section 4: consider giving a short overview of the result sections to follow, I got a bit lost in the many subsections…

We propose the following paragraph at the beginning of the Results section:

l. 308 (Results): "***The Results section is divided into three parts. In the first part, we present the ROS events that took place at the MF and BRV sites, along with the snowpack observations at both sites. In the second part, we describe the evaluation of the SNOWPACK model using the IM version of the canopy module. In the third part, we present the effects of implementing the ISD parameterization into SNOWPACK on snow unloading, snowpack structure and snowpack runoff during ROS events, along with a case study of a specific ROS event and the sensitivity analysis of the ISD function parameters.***"

**25.** Figure 3 is nice, but it would help a lot to add the acquisition times of snow pits. This would also help interpret the information in Table 4, because I guess you are much more certain about the formation date of layers where you have more frequent observations? And the number of observations is also directly related to number of field visits.

This is a good idea. The acquisition dates of snow pits will be added to Figure 3 in the revised version of the manuscript and the caption will be modified accordingly.

**26.** Table 4: 'The formation date is assumed to be the first rain-on-snow event since the previous layer was observed for the first time.' This took me multiple reads to digest. Suggest changing to: 'The formation date is assumed to be the date of the first rain-on-snow event that occurred after the date of the snow pit acquisition during which the layer beneath the melt-freeze layer was observed for the first time.' (I know it's bulky…)

Although the text is longer, it makes more sense to write it this way. We will change the table note for the one suggested above.

**27.** Figure 5: Really cool pictures. For d): what made you think that these water pockets formed in the snowpack, rather than being chunks of unloading snow? Same for e)/f)?

Except for the presence of liquid water, there was no clear discontinuity between the snow identified as "water pockets" and the surrounding snow. For e) and f), the icy structures showed a clear vertical shape of percolation channels. We suggest adding the following clarifications:

l.352 to 353 (Results): "*The absence of a continuous vertical wetting between these pockets suggests that water was transported downward by preferential flow. **Also, since there was no discontinuity between***

*the snow identified as "water pockets" and the surrounding snow, we believe that these were pockets of liquid water rather than pieces of unloaded snow.*"

l. 354 to 355 (Results): "*These ice structures, **which show a clear vertical shape,** appear to be residual preferential channels melting at slower rates than the surrounding snow, as also reported by Teich et al. (2019)."*

**28.** Figure 6: 'The observed temperature was forced to a maximum of 0°C': Do you mean that positive values were set to 0? Are these noise or a constant bias?

We did not observe a constant bias of the temperature sensors above 0°C. We forced positive temperatures to 0°C to correct for the noise.

**29.** Figure 7 and potentially 8: Why did you decide not to include the ISD simulation in these Figures? It should be straightforward and would help contextualize the later results (without making the figures too dense)

ISD simulations were very similar to IM simulations for snow height and snow surface temperature time series. We could barely differentiate one curve from the other on the same figure.

**30.** Figure 10a: It seems that between November and mid-March, unloading in the model happens mainly in three big events, the two first of which match time periods in which the time lapse cameras detect the canopy becoming snow free. If I understand the way unloading is treated in the model, these large events can only be triggered by sudden large change in interception capacity and would coincide with ROS events. If that's correct, please mention it explicitly, it would really help interpretation of the results.

We suggest the following addition in the manuscript:

l. 428 (Results): "***The model reproduces well the observations of the canopy becoming snow-free due to the large drop in interception capacity triggered by ROS events, leading to large unloading events.***"

**31.** Figure 10c: Looking at the other density profiles in the Supplementary Material, this seems to be the only case where the impact of ISD on surface density was observed. So it might be useful to explicitly point at the 'layers resulting from unloading' mentioned in L 442.

We suggest the following addition:

l. 441 (Results): "*...which strongly underestimates density **resulting from snow unloading**.*"

**32.** L 444 ff (to end of section): Really nice result!

Thank you!

**33.** L 505: Strictly speaking, this is just the model sensitivity, you don't have experimental evidence. I would remove this statement / integrate with the previous sentence.

We agree that this statement should be modified. However, the fact that we influence snowpack runoff from ROS events by modifying canopy snow parameters is an interesting finding that should be explicitly mentioned. Therefore, we suggest the following changes:

l. 505 to 506 (Results): **"*Most importantly, **this** suggests that **canopy snow metamorphism may influence** runoff generation during rain-on-snow events. **Simulations in forest gaps and observations of snowpack runoff would be needed to confirm this hypothesis.*"**

**34.** L 517: Awkward sentence. Do you mean 'simulated runoff is more sensitive to canopy snow parameters than the number of melt-freeze layers'? Please revisit.

This is what we meant. We suggest changing the original sentence this way:

l. 517 (Results): "*As shown in Fig. 14, **simulated runoff appears to be more sensitive to a change in canopy snow parameters than the number of modeled melt-freeze layers.***"

**35.** L 539: 'Air temperature and amount of precipitation during a ROS event do not appear to be good predictors of melt–freeze layer thickness': Please link to the result section / figure that backs up this statement, I wasn't sure what you are alluding to.

In fact, this statement is not clear and does not provide readers with a relevant interpretation. We suggest deleting it and replacing the beginning of the paragraph with the following:

l. 538 to 542 (Discussion): "***The thickness of the melt-freeze layers from ROS events shows a high variability (Table 4). This could be attributed to the amount of precipitation and air temperature during a ROS event, as well as** the unloading of snow clumps that **are** further redistributed unevenly across the snow surface.*"

**36.** L 558: The 'rapid runoff response' would better match the results you obtained with IM – so this statement is a bit confusing.

We suggest removing this statement.

**37.** L 561: 'General overestimation of snow density' – where do you see that? The information from S8 to S15 is somewhat hard to digest...

In fact, except for winter 2028-19 at MF, the simulated snow density profile is larger than the observed profile 19 out of 22 times. However, we agree that this analysis is unnecessary for the storyline of the article. We therefore suggest removing the sentence from l. 560 to 562.

**38.** L 564: I am not sure that this is because storage capacity is reached, it might just be too small interception (maybe I am missing something?).

The simulated evaporation/sublimation from the canopy is $\approx$70% lower than what is generally observed at the MF site, which contributes to an underestimation of interception. Since the model simulates too little interception, more snow reaches the ground and the snow height is overestimated. We suggest the following modification to the manuscript:

L. 563 to 565 (Discussion): "***The overestimation of snow height** could also be explained by too little interception simulated by SNOWPACK due to an underestimation of evaporation and sublimation by the model.*"

**39.** L 570ff: Confusing; if SNOWPACK simulates too little radiation reaching the ground, it would rather UNDERestimate surface temperature? And why would the underestimation of snow surface layer density only impact the nighttime? To me this looks more like a longwave radiation effect, but I might be misunderstanding your statement. Please doublecheck.

This is a good observation. Following your remark, we have noticed that Gouttevin et al. (2015) had the problem of overestimating longwave radiation during the day and underestimating it during the night with the two-layer canopy model. Based on this, we have modified our interpretation of the results, as longwave radiation may be in fault. Here are the changes we propose to the manuscript:

l.570 to 574 (Discussion): "***The surface temperature simulated by SNOWPACK is overestimated during the day and underestimated during the night under*** *clear sky conditions.* ***This could be explained by too much and too little downwelling longwave radiation under the canopy during the day and night, respectively, contributing to snow surface warming, as noted by Gouttevin et al. (2015) for simulations at a subalpine site in Switzerland with a LAI of 3.9 $m^2$ $m^{-2}$.*** *Under cloudy conditions, SNOWPACK simulations of $T_{surf}$ improve because and because air and snow surface temperatures are similar.*"

**40.** L 619: Here it would be fair to acknowledge that some multi-layer snow models ARE already able to resolve tree-scale processes (SNOWPALM, ; FSM2, https://doi.org/10.1029/2020WR027572) – these models do not include microstructure, but concepts used therein could in principle be applied to SNOWPACK as well in the future. In this context, this preprint might be of interest as well: https://doi.org/10.5194/egusphere-2023-2781.

This is a good point. Unfortunately, the preprint from Mazzotti et al. (2023) was not yet available in the EGUSPHERE discussion when we submitted our manuscript. We now include it in this manuscript. We suggest adding the following text at the end of the paragraph:

l. 619 (Discussion): "*Therefore, further modeling developments are needed to better represent spatially variable vegetation–snow processes like interception and unloading (Vincent et al., 2018) or even radiation transfer (Jonas et al., 2020) in multi–layer,* ***microstructure resolving*** *snow models.* ***This can be achieved by coupling these detailed snow models with models that resolve tree-scale processes. Recent work by Mazzotti et al. (2023) who coupled FSM2 (Mazzotti et al., 2020) to CROCUS (Vionnet et al., 2012) looks promising to this end.***"

**41.** L 670: 'Solid or liquid unload': shouldn't this be 'unloading'?

Thank you. We will correct this.

**42.** Supplementary Material: I noticed that the ISD simulation in figure S20 onwards is labelled 'Snow Tracking' – maybe from an older version? It's a detail and only the Supplementary Material, but consider correcting this for consistency.

Indeed, this is the previous name of our model development. Figures S20 to S23 have not been updated with the most recent name (Int. Snow Dens.). This will be corrected in the revised version.

DOI of the newly added references:

- Andreadis et al. (2009) [doi: 10.1029/2008WR007042]
- Calonne et al. (2012) [doi: 10.5194/tc-6-939-2012]
- Fang et al. (2019) [doi: 10.1029/2018RG000608]
- Mazzotti et al. (2020) [doi : 10.1029/2020WR027572]
- Mazzotti et al. (2023) [doi: 10.5194/egusphere-2023-2781]
- Musselman et al. (2008) [doi: 10.1002/hyp.7050]
- Roesch et al (2001) [doi: 10.1007/s003820100153]
- Vionnet et al. (2012) [doi: 10.5194/gmd-5-773-2012]